# The quantitative architecture of centromeric chromatin

**Dani L Bodor[1], João F Mata[1], Mikhail Sergeev[2,3], Ana Filipa David[1], Kevan J Salimian[4], Tanya Panchenko[4], Don W Cleveland[5,6], Ben E Black[4], Jagesh V Shah[2,3], Lars ET Jansen[1]***

[1]Instituto Gulbenkian de Ciência, Oeiras, Portugal; [2]Department of Systems Biology, Harvard Medical School, Boston, United States; [3]Renal Division, Brigham and Women's Hospital, Boston, United States; [4]Department of Biochemistry and Biophysics, Perelman School of Medicine, University of Pennsylvania, Philadelphia, United States; [5]Ludwig Institute for Cancer Research, University of California, San Diego, La Jolla, United States; [6]Department of Cellular and Molecular Medicine, University of California, San Diego, La Jolla, United States

**Abstract** The centromere, responsible for chromosome segregation during mitosis, is epigenetically defined by CENP-A containing chromatin. The amount of centromeric CENP-A has direct implications for both the architecture and epigenetic inheritance of centromeres. Using complementary strategies, we determined that typical human centromeres contain ~400 molecules of CENP-A, which is controlled by a mass-action mechanism. This number, despite representing only ~4% of all centromeric nucleosomes, forms a ~50-fold enrichment to the overall genome. In addition, although pre-assembled CENP-A is randomly segregated during cell division, this amount of CENP-A is sufficient to prevent stochastic loss of centromere function and identity. Finally, we produced a statistical map of CENP-A occupancy at a human neocentromere and identified nucleosome positions that feature CENP-A in a majority of cells. In summary, we present a quantitative view of the centromere that provides a mechanistic framework for both robust epigenetic inheritance of centromeres and the paucity of neocentromere formation.

*For correspondence: ljansen@igc.gulbenkian.pt

**Competing interests:** The authors declare that no competing interests exist.

**Reviewing editor**: Jon Pines, The Gurdon Institute, United Kingdom

## Introduction

Centromeres are essential for proper cell division. During mitosis, a transient structure called the kinetochore is assembled onto centromeric chromatin, which mediates the interaction between DNA and the mitotic spindle (*Allshire and Karpen, 2008*; *Cheeseman and Desai, 2008*). Intriguingly, although centromeres are directly embedded in chromatin, specific DNA sequences are neither necessary nor sufficient for centromere function. This is best exemplified by the rare occurrence, within the human population, of neocentromeres: functional centromeres that have repositioned to atypical loci on the chromosome (*Amor et al., 2004*; *Marshall et al., 2008*; *du Sart et al., 1997*; *Voullaire et al., 1993*). Rather than centromeric sequences, the primary candidate for epigenetic specification of centromeres is the histone variant CENP-A, which replaces canonical H3 in centromeric nucleosomes (*Palmer et al., 1987*, *1991*; *Stoler et al., 1995*; *Henikoff et al., 2000*; *Yoda et al., 2000*). CENP-A chromatin is sufficient for recruitment of the downstream centromere and kinetochore complexes (*Foltz et al., 2006*; *Okada et al., 2006*; *Carroll et al., 2009*, *2010*; *Barnhart et al., 2011*; *Guse et al., 2011*; *Mendiburo et al., 2011*). In addition, CENP-A is stably transmitted at centromeres during mitotic (*Jansen et al., 2007*; *Bodor et al., 2013*) and meiotic (*Raychaudhuri et al., 2012*) divisions, and its assembly is tightly cell cycle controlled (*Jansen et al., 2007*; *Schuh et al., 2007*; *Silva et al., 2012*). Importantly, targeting of this protein to an ectopic site of the genome is sufficient to initiate an epigenetic feedback loop,

**eLife digest** The genetic information in a cell is packed into structures called chromosomes. These contain strands of DNA wrapped around proteins called histones, which helps the long DNA chains to fit inside the relatively small nucleus of the cell.

When a cell divides, it is important that both of the new cells contain all of the genetic information found in the parent cell. Therefore, the chromosomes duplicate during cell division, with the two copies held together at a single region of the chromosome called the centromere. The centromere then recruits and coordinates the molecular machinery that separates the two copies into different cells.

Centromeres are inherited in an epigenetic manner. This means that there is no specific DNA sequence that defines the location of this structure on the chromosomes. Rather, a special type of histone, called CENP-A, is involved in defining its location. Bodor et al. use multiple techniques to show that human centromeres normally contain around 400 molecules of CENP-A, and that this number is crucial for ensuring that centromeres form in the right place. Interestingly, only a minority of the CENP-A molecules are located at centromeres; yet this is more than at any other region of the chromosome. This explains why centromeres are only formed at a single position on each chromosome.

When the chromosomes separate, the CENP-A molecules at the centromere are randomly divided between the two copies. In this way memory of the centromere location is maintained. If the number of copies of CENP-A inherited by one of the chromosomes drops below a threshold value, a centromere will not form. However, Bodor et al. found that the number of CENP-A molecules in a centromere is large enough, not only to support the formation of the centromere structure, but also to keep it above the threshold value in nearly all cases. This threshold is also high enough to make it unlikely that a centromere will form in the wrong place because of a random fluctuation in the number of CENP-A molecules. Therefore, the number of CENP-A molecules is crucial for controlling both the formation and the inheritance of the centromere.

recruiting more CENP-A to this site (*Mendiburo et al., 2011*). However, little is known about the quantity of CENP-A present at centromeres, despite this being an essential parameter for a functional understanding of both centromeric architecture and epigenetic inheritance. Here, we use multiple, independent approaches to determine the absolute copy number of CENP-A at centromeres. In addition, we provide novel insights in the mechanisms of centromere size control.

## Results

### Modification of endogenous CENP-A alleles in diploid human cells

To determine absolute centromeric CENP-A levels in human cells, we set out to build cell lines in which the entire CENP-A pool is fluorescent. To accomplish this, we removed a significant and essential portion of the CENP-A gene to create a knock-out allele in stably diploid, human retinal pigment epithelium (RPE) cells (*Figure 1A*, bottom). Subsequently, a fluorescent knock-in allele was created by placing GFP or YFP encoding sequences in frame with the sole remaining CENP-A gene (*Figure 1A*, middle). Specifically, we have built the following endogenously targeted RPE cell lines: CA$^{+/-}$, CA$^{G/-}$, CA$^{Y/-}$, and CA$^{+/F}$ (where + = wild-type; − = knock-out; G = GFP knock-in; Y = YFP knock-in; F = floxed [to control for potential gene-targeting artifacts]; *Figure 1—figure supplement 1A*). Western blot analysis confirms that CA$^{G/-}$ and CA$^{Y/-}$ cells exclusively contain tagged CENP-A (of ~43 kDa), while CA$^{+/+}$ (wild-type), CA$^{+/F}$, and CA$^{+/-}$ cells only express wild-type CENP-A (~16 kDa) protein (*Figure 1B*). Importantly, heterozygous expression or tagging of endogenous loci did not interfere with cell viability.

### Centromeric CENP-A levels are regulated by mass-action

While CENP-A is an essential and constitutive component of centromeres, how the size of the centromeric chromatin domain is controlled is not known. We analyzed the consequences of different CENP-A expression levels in our CENP-A heterozygous knock-out and knock-in lines, as well as in a cell line that ectopically overexpressed CENP-A-YFP (CA$^{Y/-}$+OE; *Figure 1B; Figure 1—figure supplement 1A*).

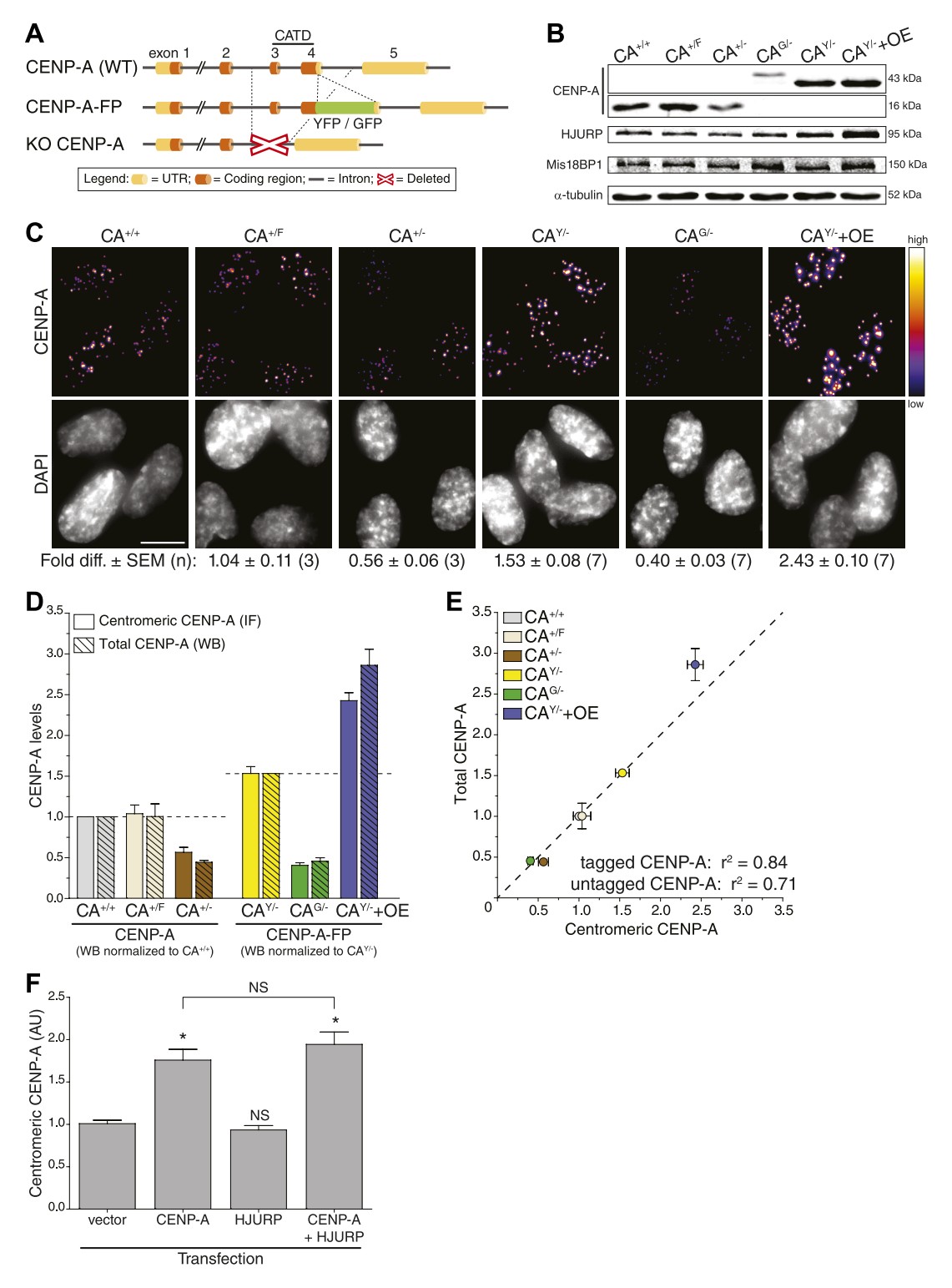

**Figure 1**. CENP-A levels are regulated by mass-action. (**A**) Schematic of gene-targeting strategy that allowed for the creation of CENP-A knockout and fluorescent knock-in alleles. The region encoding the essential CENP-A targeting domain (CATD, **Black et al., 2007**) is indicated. (**B**) Quantitative immunoblots of CENP-A, HJURP, and Mis18BP1 in differentially targeted RPE cell lines. α-tubulin is used as a loading control. (**C**) Immunofluorescence images of same cell lines as in **B**. CENP-A intensity is represented in a heat map as indicated on the right. The fold difference ± SEM (n is biological

*Figure 1. Continued on next page*

*Figure 1. Continued*

replicates) compared to wild-type RPE cells is indicated below. Scale bar: 10 μm. Note that in contrast to quantification of immunoblots, immunofluoresce detection of untagged and tagged CENP-A is directly comparable. (**D**) Quantification of centromeric CENP-A levels (from **C**) by immunofluorescence (IF) and total CENP-A levels (n = 4–9 independent experiments as in **B**) by western blot (WB). All cell lines expressing untagged CENP-A are normalized to CA$^{+/+}$ while those expressing tagged CENP-A are normalized to the centromeric CA$^{Y/-}$ levels measured in **C**, as indicated by dashed lines. (**E**) Correlation of centromeric and total cellular CENP-A levels as measured in **D**. Dashed line represents a predicted directly proportional relationship with indicated correlation coefficients. Throughout, the average ± SEM is indicated. (**F**) Quantification of centromeric CENP-A levels in synchronized HeLa cells (based on anti-CENP-A staining) within a single cell cycle after transient transfection of indicated proteins. Asterisk indicates statistically significant increase compared to control or indicated transfections (one-tailed *t* test; p<0.05; n = 3); NS indicates no significant increase. Average ± SEM of three independent experiments is shown.

The following figure supplements are available for figure 1:

**Figure supplement 1**. CENP-A expression is the rate limiting factor for centromeric CENP-A levels.

First, we measured the total protein pool of CENP-A in our cell lines by quantitative immunoblotting. While we found the detection output for CENP-A to be linear over at least a 32-fold range (*Figure 2E*), due to differences in protein transfer efficiencies this method does not allow for a comparison between proteins of different sizes, for example (GFP- or YFP-) tagged and untagged (wild-type) CENP-A (*Figure 2—figure supplement 3*). Nevertheless, we could directly compare CA$^{G/-}$, CA$^{Y/-}$, and CA$^{Y/-}$+OE cell lines (*Figure 2—figure supplement 3*) and found that cellular CENP-A content spans a sixfold range (*Figure 1B,D*).

Given its essential role in centromere function, we predicted a tight control of centromeric CENP-A levels. However, instead of maintaining a fixed amount of CENP-A at centromeres, the levels varied extensively (*Figure 1C*). Both CA$^{+/-}$ and CA$^{G/-}$ cells, which contain a single intact allele, have decreased centromeric CENP-A levels, while the parental CA$^{+/F}$ cells maintain wild-type levels. Surprisingly, despite expressing CENP-A from a single allele, CA$^{Y/-}$ cells have increased CENP-A levels, which may be due to adaptations that arose during the creation of this cell line. As expected, CENP-A levels are further elevated in CA$^{Y/-}$+OE cells (*Figure 1C*). Remarkably, we found a very high correlation (r$^2$ = 84%) for a hypothetical directly proportional relationship between centromeric and total cellular CENP-A-GFP or CENP-A-YFP levels (*Figure 1D,E*). Similarly, despite an only approximately twofold range of expression, we still observe a high correlation with direct proportionality (r$^2$ = 71%) for cells expressing untagged CENP-A (*Figure 1D,E*). Thus, our observations indicate that centromeric levels are determined by a mass-action mechanism, where the amount of centromeric CENP-A varies in direct proportion with the cellular content.

An alternative hypothesis is that stable cell lines have undergone long-term adaptation to altered CENP-A expression, which has led to re-equilibrated centromeric levels. For example, proteins involved in depositing CENP-A at the centromere may have adapted to CENP-A expression levels. Indeed, we see a weak correlation between the levels of CENP-A and its histone chaperone HJURP (*Dunleavy et al., 2009*; *Foltz et al., 2009*; *Barnhart et al., 2011*) in our cell lines (*Figure 1B*, *Figure 1—figure supplement 1B*). Conversely, no correlation was detected for Mis18BP1 (*Figure 1B*, *Figure 1—figure supplement 1C*), another essential protein for CENP-A assembly (*Fujita et al., 2007*; *Maddox et al., 2007*), arguing that it is a non-stoichiometric component of the loading pathway. To test whether centromeric CENP-A levels require long-term adaptation, we analyzed the effect of CENP-A and/or HJURP overexpression in a single round of CENP-A assembly. Therefore, we transiently expressed CENP-A and/or HJURP and measured the level of centromeric CENP-A after a single cell cycle in HeLa cells, which can be effectively synchronized in S phase using thymidine. While induction of CENP-A expression leads to a prompt increase in centromeric levels, no (additional) effect was observed by expression of HJURP (*Figure 1F*). Together, our results strongly suggest that centromeric CENP-A levels are directly regulated by its protein expression levels.

## Centromeres contain ~400 molecules of CENP-A

To understand how CENP-A chromatin is self-propagated and nucleating the kinetochore, it is critical to establish the absolute amount of CENP-A present. In vertebrates, previous estimates range from a few tens of molecules (in chicken DT40 cells, *Ribeiro et al., 2010*) to a potential maximum of tens of thousands (in HeLa cells, *Black et al., 2007*). To directly determine the copy number of CENP-A on

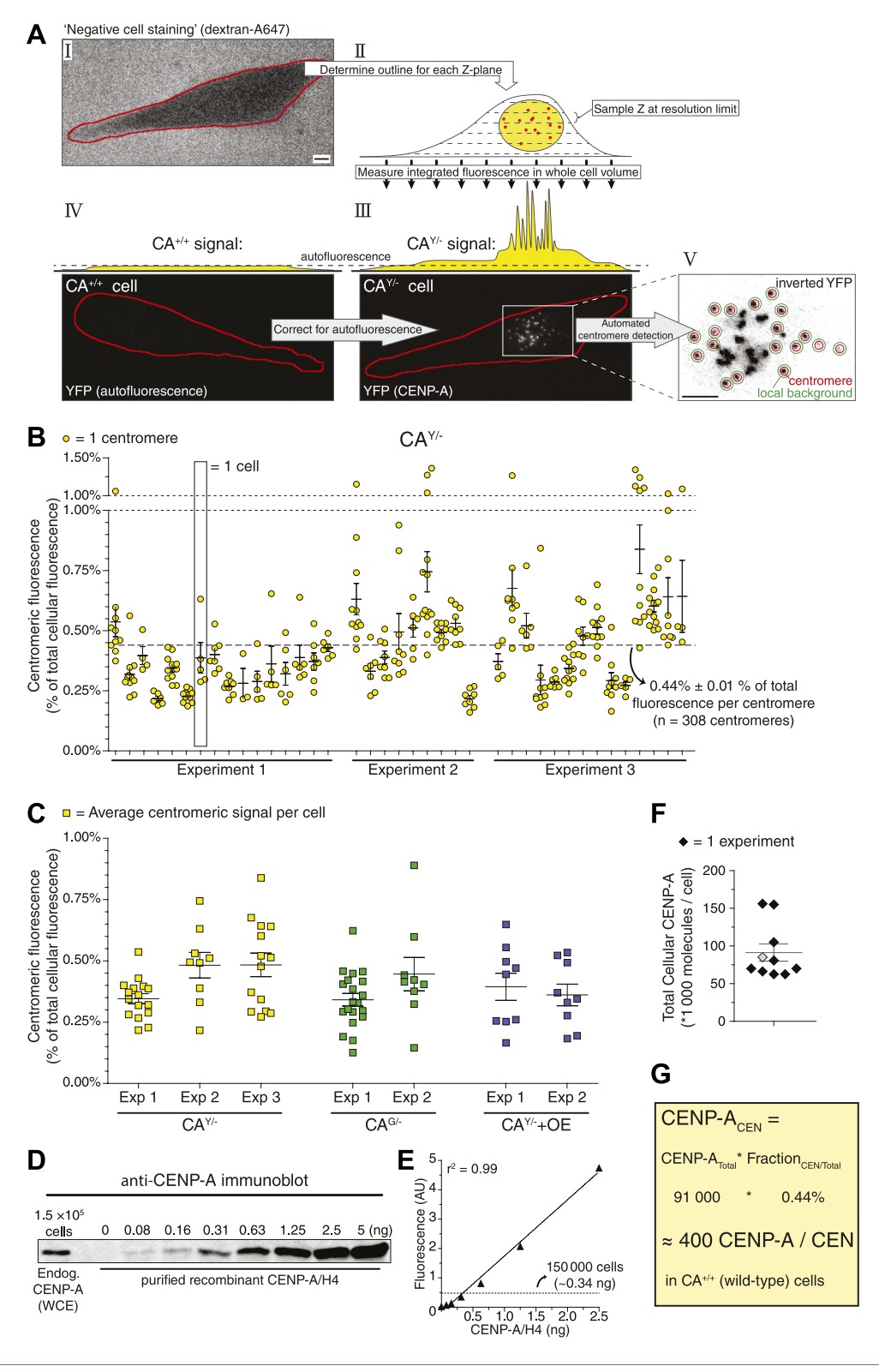

**Figure 2**. Human centromeres contain 400 molecules of CENP-A. (**A**) Schematic outline of strategy allowing for the quantification of the centromeric fraction of CENP-A compared to the total cellular pool. Scale bars: 5 μm. (**B**) Quantification of the centromeric fraction of CENP-A in CA^Y/− cells. Each circle represents one centromere; *Figure 2. Continued on next page*

*Figure 2. Continued*

circles on the same column are individual centromeres from the same cell. Dashed line indicates average of all centromeres. (**C**) Quantification of the centromeric fraction of CENP-A in indicated cell lines. Each square represents the average centromeric signal from one cell; squares on the same column are individual cells from the same experiment (Exp). *Figure 2—figure supplement 2* shows quantification of individual centromeres in CA$^{G/-}$ and CA$^{Y/-}$+OE cells. (**D**) Representative quantitative immunoblot of purified recombinant CENP-A and endogenous CENP-A from whole cell extracts (WCE). (**E**) Quantification of **D**. Solid line represents the best fit linear regression. Dashed line represents the amount of CENP-A from 150,000 cells. (**F**) Quantification of the total cellular CENP-A copy number. Each diamond represents one replicate experiment; measurement from **E** is indicated as a gray diamond. (**G**) Calculation of average CENP-A copy number per centromere (CEN) in wild-type RPE cells. Throughout, the average ± SEM is indicated.

The following figure supplements are available for figure 2:

**Figure supplement 1**. Representative fluorescence lifetime imaging (FLIM) micrograph of a CENP-A-YFP expressing cell (left) and quantification of indicated cellular regions (right).

**Figure supplement 2**. Measurements of individual centromeres and CENP-A levels for different cell lines.

**Figure supplement 3**. Transfer efficiency of recombinant and cellular CENP-A.

---

human centromeres, we developed a 3D imaging strategy (*Figure 2A*), which we adapted from a method used previously to quantify cytokinesis proteins in fission yeast (*Wu and Pollard, 2005*; *Wu et al., 2008*). In brief, we use a non-cell permeable dye (*Figure 2A*, I) to determine the 3D shape of cells (*Figure 2A*, II) and measure the total amount of fluorescence within the entire cell volume (*Figure 2A*, III). Total cellular fluorescence of CA$^{Y/-}$ cells (*Figure 2A*, III) was corrected for autofluorescence measured in wild-type RPE cells (*Figure 2A*, IV), thus resulting in a measure of total CENP-A-derived fluorescence. Next, centromere-specific fluorescence was measured after correction for local background (*Figure 2A*, V; *Hoffman et al., 2001*) and axial oversampling. Importantly, fluorescence lifetime of CENP-A-YFP is similar between highly concentrated centromeric and diffuse cytoplasmic pools (*Figure 2—figure supplement 1*), arguing that clustering does not lead to changes in fluorescence efficiency. In effect, our 3D-integrated fluorescence strategy measures the centromeric fraction of CENP-A compared to the total cellular pool. We find that while CENP-A is enriched at centromeres, on average only 0.44% of cellular CENP-A is present per centromere in CA$^{Y/-}$ cells (*Figure 2B*). Very similar fractions were observed in CA$^{G/-}$ and CA$^{Y/-}$+OE cells (0.38% in both cases; *Figure 2C*, *Figure 2—figure supplement 2A,B*), which provides an additional line of evidence in support of a mass-action mechanism for CENP-A assembly. Furthermore, these findings show that a surprising minority, about one-fifth of the CENP-A protein content (0.44% × 46) is present on the functionally relevant subcellular location, i.e. at the centromeres.

To convert centromeric fractions to absolute amounts, we determined the total number of CENP-A molecules in RPE cells. To this end, we prepared whole cell extracts of RPE cells and analyzed these alongside highly purified recombinant CENP-A/H4-complexes of known concentration by quantitative immunoblotting (*Figure 2D*). Importantly, we ensured that recombinant and cellular CENP-A have the same transfer efficiency and can be directly compared to each other (*Figure 2—figure supplement 3*). Fitting the cellular amount of CENP-A onto a linear regression curve of purified protein (*Figure 2E*) shows that CA$^{+/+}$ cells contain an average of ~9.1 ± 1.1 × 10$^4$ (n = 10) molecules of CENP-A per cell (*Figure 2F*). Because the centromeric fraction of CENP-A is fixed, we can calculate the absolute amount of CENP-A per centromere in our cell lines (*Figure 2G*, *Figure 2—figure supplement 2C*) and show that wild-type RPE cells contain ~400 molecules of CENP-A on an average centromere.

Both the expression and centromeric loading of CENP-A are cell cycle regulated (*Figure 3A*). In human cells, cellular protein levels of CENP-A peak in late G2 (*Shelby et al., 2000*), while incorporation into centromeric chromatin occurs in early G1 phase (*Jansen et al., 2007*). Thus, it is possible that part of the cell-to-cell variation of the centromeric CENP-A ratio observed in *Figure 2C* is due to differing cell cycle stages. We tested this possibility using a previously developed fluorescent ubiquitin-based cell cycle indicator (FUCCI) that can be used in live cells (*Sakaue-Sawano et al., 2008*). In particular, we used hCdt1(30/120)-RFP, which is expressed ubiquitously throughout the cell cycle, but

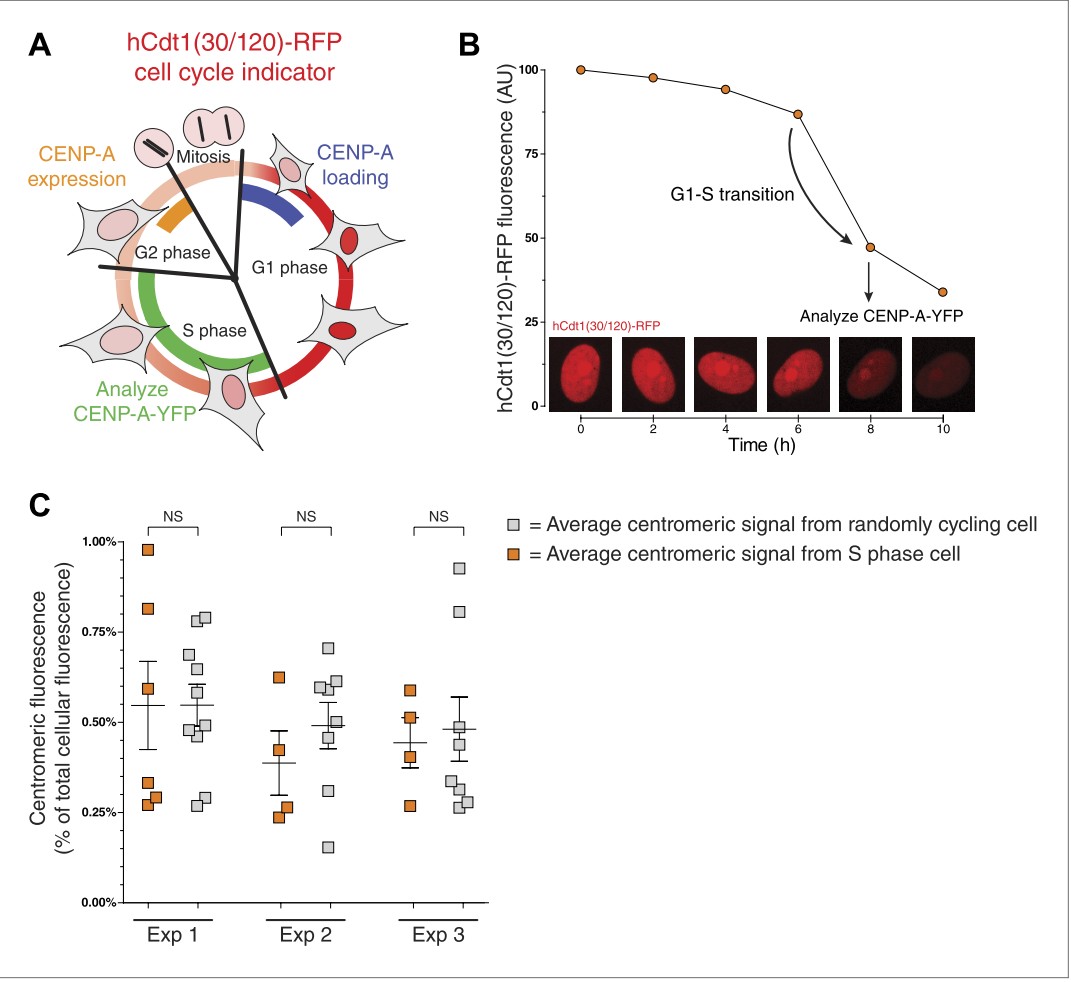

**Figure 3**. Centromeric CENP-A levels are equivalent between S phase and randomly cycling cells. (**A**) Cartoon depicting changes in cell morphology and nuclear levels of hCdt1(30/120)-RFP (in red) throughout the cell cycle (*Sakaue-Sawano et al., 2008*). Approximate timing of CENP-A expression (*Shelby et al., 2000*) and centromeric loading (*Jansen et al., 2007*) are indicated in orange and blue, respectively. The stage at which cells were analyzed to measure the centromeric fraction of CENP-A is indicated in green. (**B**) An example trace of a cell entering S phase (indicated by a sudden decrease in RFP levels) is shown. The centromeric fraction of CENP-A was measured at this point as outlined in *Figure 2A*. Peak expression is normalized to 100 and background fluorescence to 0. Micrographs of hCdt-1(30/120)-RFP at indicated timepoints are shown below. (**C**) As in *Figure 2C*. Orange squares represent cells that have passed the G1-S transition point, as indicated by decreasing levels of hCdt-1(30/120)-RFP. Gray squares represent randomly cycling cells. No statistically significant differences (NS) were observed between randomly cycling cells and S phase cells.

The following figure supplements are available for figure 3:

**Figure supplement 1**. hCdt-1(30/120)-RFP expression allows for accurate determination of cell cycle stages and measurements of centromeric CENP-A ratios.

is specifically degraded in S, G2, and M phases (*Sakaue-Sawano et al., 2008*). As a result, protein levels increase as cells enter and progress through G1 phase, peak at the G1/S boundary, and then drop until cells re-enter G1 (*Figure 3A*). We expressed this protein in CA$^{Y/-}$ cells and tracked the RFP fluorescence intensity over time (*Figure 3B*, *Figure 3—figure supplement 1A*) to identify cells that entered S phase (see 'Materials and methods' for details). We compared the centromeric fraction of CENP-A of S Phase cells to that of randomly cycling cells and found that neither the mean nor the variance differs significantly between these two populations (*Figure 3C*). Importantly, expression of the FUCCI marker itself has no effect on the measurements performed (*Figure 3—figure supplement 1B*).

While the centromeric fraction of CENP-A is likely low in G2 phase and high just after assembly in early G1, we find that the variation observed in *Figure 2C* is not a consequence of such cell cycle-induced effects and may instead reflect inherent variation between cells.

Although the method we employed to measure centromeric ratios is internally controlled, it relies on measurement of integrated fluorescence of whole cells, including highly dilute cytoplasmic CENP-A. To exclude potential errors in measurements of low protein concentration, we stably expressed H2B-RFP in CA^{Y/−} cells (*Figure 4A*, inset) and determined that 0.73% of nuclear CENP-A is present on each centromere (*Figure 4A*). In addition, low salt fractionation experiments indicate that ~74% of cellular CENP-A co-pellets with other chromatin components in CA^{Y/−}+H2B-RFP cells (*Figure 4B*), indicating

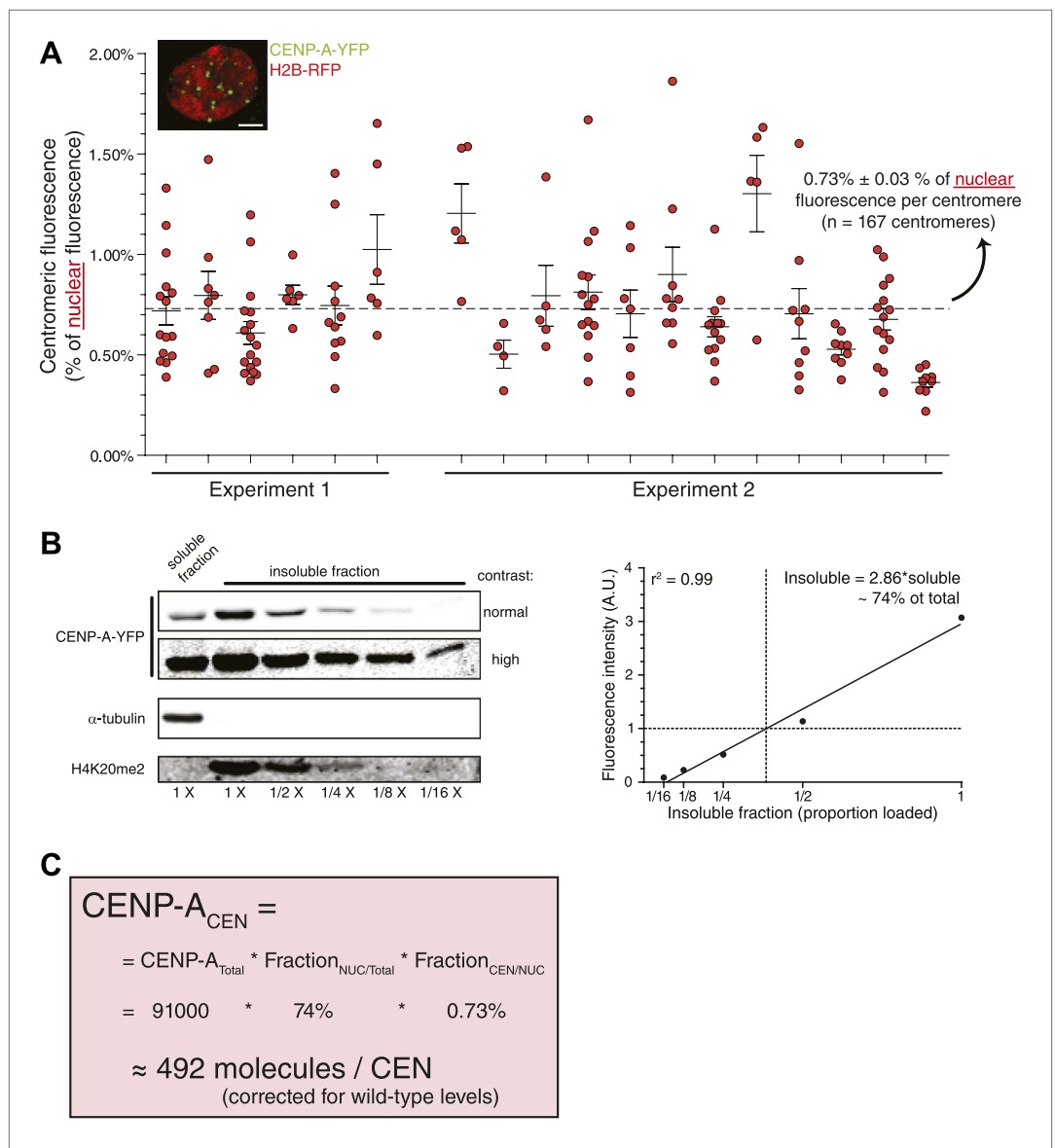

**Figure 4**. Measurement of nuclear CENP-A confirms centromeric copy number. (**A**) As in *Figure 2B*, except that the centromeric fraction compared to total nuclear pool is indicated. Inset shows a representative image of a CA^{Y/−}+H2B-RFP cell (scale bar: 2.5 μm). (**B**) Quantitative immunoblot showing the soluble fraction and a dilution series from the insoluble fraction of CENP-A-YFP in CA^{Y/−}+H2B-RFP cells (left). Tubulin is used as a marker for the soluble fraction and H4K20me2 (exclusively found in chromatin, *Karachentsev et al., 2007*) for the insoluble fraction. Quantification of insoluble fraction of CENP-A is shown to the right. (**C**) Calculation of the average CENP-A copy number per centromere (CEN) in wild-type RPE cells, based on results from CA^{Y/−}+H2B-RFP cells.

that this represents the stable nuclear pool. Combined, we find a similar number of CENP-A molecules per centromere when analyzing the nuclear pool (492 molecules; *Figure 4C*) as when measuring total cellular CENP-A. This argues that the measurements performed above are not significantly influenced by a potential inaccuracy in determining the cytoplasmic pool. Interestingly, it has recently been shown that detectable levels of CENP-A are assembled into non-centromeric chromatin of HeLa cells (*Lacoste et al., 2014*). We now find that, at least in RPE cells the proporation of chromatin bound CENP-A outside of the centromere is surprisingly high (~66% in this cell line).

## Centromeric CENP-A copy number is confirmed by three independent methods

To further validate that the strategy described above accurately measures centromeric CENP-A copy numbers, we used two additional independent quantification methods. First, we applied a method that employs the statistical properties of fluorescence redistribution (*Rosenfeld et al., 2005*, *2006*). This method relies on the fact that random segregation of fluorescent molecules leads to each daughter receiving an (unequal) fraction, where the distribution of differences relates to the total number of molecules (as outlined in *Figure 5A*). During mitosis, sister centromeres form individually resolved spots by light microscopy, allowing us to measure the fluorescence intensity of individual sisters (*Figure 5B*). We find that rather than accurately segregating exactly half of pre-assembled CENP-A onto each daughter chromatid, the difference between sister centromeres follows a random distribution (*Figure 5B,C*). Previously, Rosenfeld et al. have provided mathematical evidence that measurements of this deviation allow for the determination of the fluorescence intensity of a single heritable, segregating unit (*Figure 5A*, *Rosenfeld et al., 2005*, *2006*). We measured an average of 75.4 segregating units of CENP-A-GFP per centromere in CA$^{G/-}$ cells (*Figure 5D*). Because each segregating unit consists of one or more nucleosomes, containing two molecules of CENP-A each (*Sekulic et al., 2010*; *Tachiwana et al., 2011*; *Bassett et al., 2012*; *Hasson et al., 2013*; *Padeganeh et al., 2013*), an average CA$^{G/-}$ centromere has a minimum of 150.8 molecules of CENP-A. Correcting the amount of CENP-A measured in CA$^{G/-}$ cells for wild-type levels (*Figure 1C*) results in ≥377 molecules of CENP-A per centromere (*Figure 5D*, right y-axis). Importantly, these measurements differ significantly if random centromere pairs are chosen for which no statistical correlation exists (*Figure 5—figure supplement 1E*). This confirms that fluorescence intensities at sister centromeres co-vary and renders this type of analysis suitable for centromere quantification. Stochastic fluctuation measurements in CA$^{Y/-}$ and CA$^{Y/-}$+OE cells indicates that wild-type cells contain ≥188 and ≥149 CENP-A molecules per centromere, respectively (*Figure 5—figure supplement 1A–D*). Importantly, the number of co-segregating CENP-A nucleosomes is unknown, which can be one or more. Therefore, despite the variation between the cell lines used here, all results obtained from this method provide a minimum estimate of the centromeric CENP-A copy number that is in agreement with the 400 centromeric molecules of CENP-A measured above (*Figure 2G*).

Next, we used a yeast strain that harbors a chromosomally integrated 4 kb LacO-array and expresses GFP-LacI as a calibrated fluorescent standard (*Lawrimore et al., 2011*). While there is a potential for 204 molecules of GFP-LacI to be bound to this array (*Lawrimore et al., 2011*), it is unlikely that the entire array is fully occupied at any moment. Because CA$^{G/-}$ cells express the same version of GFP as this yeast strain, direct comparison of fluorescent foci (*Figure 5E*) provides a maximum estimation of the centromeric CENP-A-GFP copy number. In this way, we determined that CA$^{G/-}$ centromeres contain at most 215 ± 32 CENP-A-GFP molecules, which translates to ≤538 CENP-A molecules in wild-type cells (*Figure 5F*).

Importantly, the copy number that we measure directly by our 3D integrated fluorescence approach is in close agreement with minimum and maximum estimates of the stochastic fluctuation and fluorescent standard approaches, respectively (*Figure 5G*). This provides confidence that 400 molecules of CENP-A per centromere in wild-type RPE cells is an accurate measure.

## Assessing the critical number of CENP-A nucleosomes

While cells are able to survive with a sixfold range of CENP-A levels (*Figure 1D*), centromere function may be compromised when levels are not accurately maintained. Indeed, based on a conserved ratio of centromere and kinetochore proteins and kinetochore microtubules between multiple yeast species as well as chicken DT40 cells, it has been hypothesized that centromeres form modular structures by repeating individual structural subunits (*Joglekar et al., 2008*; *Johnston et al., 2010*), as originally proposed by *Zinkowski et al. (1991)*. Thus, the amount of CENP-A would directly reflect the number

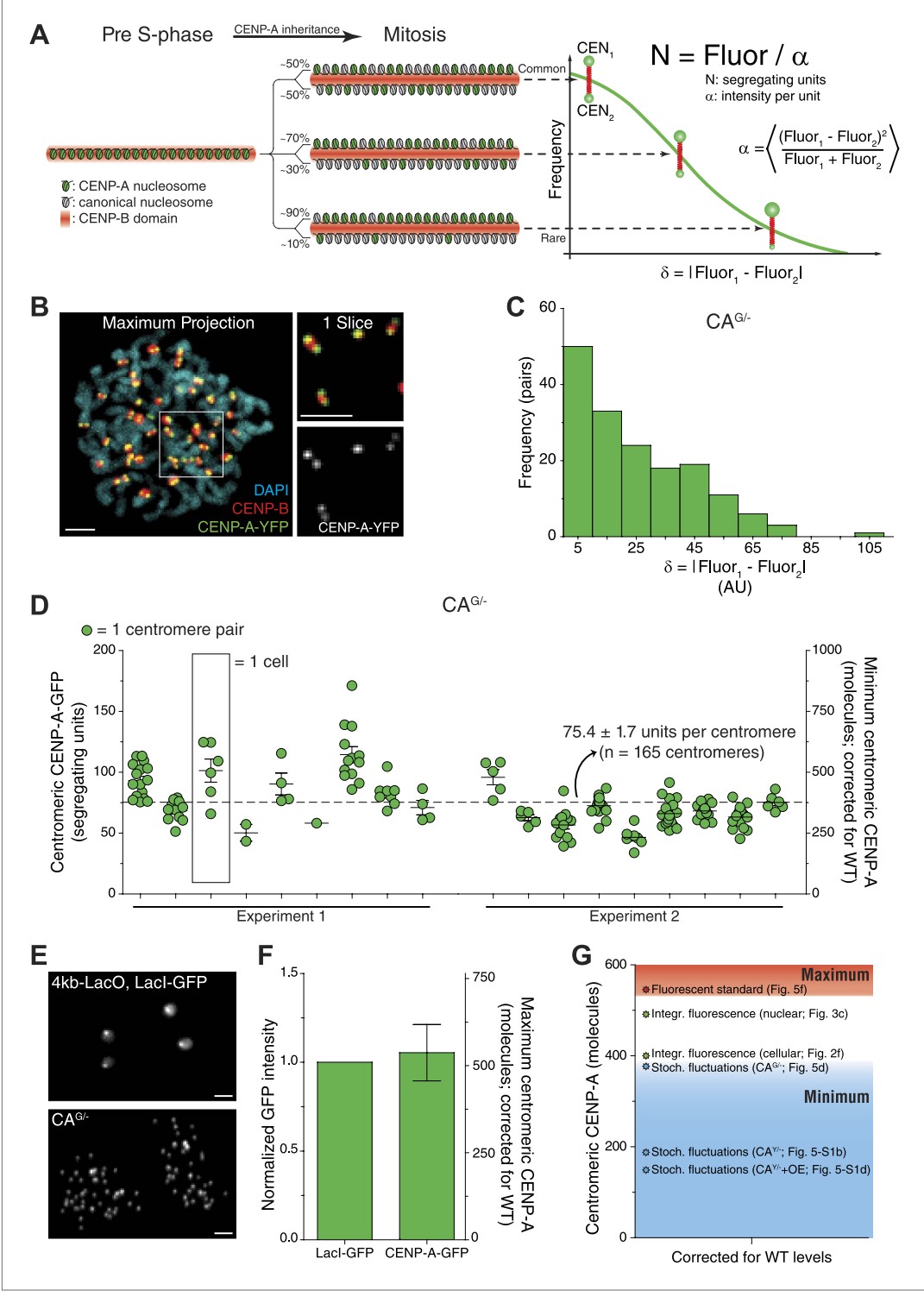

**Figure 5**. Independent quantification methods confirm centromeric CENP-A copy number. (**A**) Stochastic fluctua-tion method: cartoon depicting inheritance and random redistribution of parental CENP-A nucleosomes onto sister chromatids during DNA replication. A hypothetical distribution of the absolute difference between the two sister centromeres, as well as the formula for calculating the fluorescence intensity per segregating unit (α) are indicated on the right. (**B**) Representative image of mitotic CENP-A-YFP expressing cell. CENP-B staining allows for identifi-cation of sister centromeres. Blowup to the right represents a single slice of the boxed region showing that CENP-B is located in between the CENP-A spots of sister centromeres. (**C**) Frequency distribution of the difference between

*Figure 5. Continued on next page*

*Figure 5. Continued*

CENP-A-GFP intensity of sister centromeres in CA$^{G/-}$ cells. (**D**) Quantification of centromeric CENP-A-GFP based on the stochastic fluctuation method. Each circle represents one centromere; circles on the same column are individual centromeres from the same cell. Left y-axis indicates segregating CENP-A-GFP units in CA$^{G/-}$ cells; right y-axis shows the conversion to minimum number of centromeric CENP-A molecules in CA$^{+/+}$ (WT) cells. (**E**) Fluorescent standard method: representative fluorescence images of 4kb-LacO, LacI-GFP *S. cerevisiae* and human CA$^{G/-}$ cells. (**F**) Quantification of fluorescence signals of LacI-GFP and CENP-A-GFP spots (n = 2 biological replicates). The left y-axis indicates the fluorescence intensity normalized to LacI-GFP; the right y-axis shows the conversion to maximum number of centromeric CENP-A molecules in wild-type cells. (**G**) Comparison of independent measurements for the centromeric CENP-A copy number (corrected for CA$^{+/+}$ levels; Stoch. fluctuations = stochastic fluctuation method [*Figure 5A*]; Integr. fluorescence = integrated fluorescence method [*Figure 2A*]). Levels from all strategies are corrected for CA$^{+/+}$ (WT) levels. Throughout, the average ± SEM and scale bars of 2.5 μm are indicated.

The following figure supplements are available for figure 5:

**Figure supplement 1**. Stochastic fluctuations of CENP-A segregation allows for copy number measurements.

of downstream centromere and kinetochore proteins and microtubule attachment sites. Conversely, experiments in human cells indicate that the centromere is assembled by multiple independent sub-complexes (*Foltz et al., 2006*; *Liu et al., 2006*). Here, we analyzed whether altering the levels of CENP-A has an effect on the recruitment of other, downstream centromere or kinetochore proteins. Both CENP-C and CENP-T rely on CENP-A for their centromeric recruitment (*Régnier et al., 2005*; *Liu et al., 2006*; *Fachinetti et al., 2013*) and have recently been shown to be responsible for mitotic recruitment of the KMN network (*Gascoigne et al., 2011*), including the key microtubule binding protein Hec1/NDC80 (*Cheeseman et al., 2006*; *DeLuca et al., 2006*). Interestingly, we found that none of these three proteins were significantly affected by altering the levels of CENP-A between 40% and 240% of wild-type levels (*Figure 6A*, *Figure 6—figure supplement 1*). In line with previous findings (*Liu et al., 2006*; *Fachinetti et al., 2013*), these results argue against a modular centromere architecture where CENP-A nucleosomes form individual binding sites for downstream components. Rather, a >2½-fold excess of CENP-A appears to be present for recruitment of centromere and kinetochore complexes of fixed pool size.

Intriguingly, despite no quantitative effect on centromeric proteins, we observed that decreasing CENP-A levels leads to an increase in the fraction of cells containing micronuclei (MN; *Figure 6B*). MN often arise as a consequence of mitotic errors, such as lagging chromosomes during anaphase (*Ford et al., 1988*), breakage of anaphase bridges (*Hoffelder et al., 2004*), or multipolar mitoses (*Utani et al., 2010*). The presence of MN can be scored by DAPI staining (*Figure 6B*, bottom). A baseline fraction of 0.53% ± 0.07% (n = 4) of wild-type CA$^{+/+}$ cells contain MN (*Figure 6B*). Both cell lines that have decreased CENP-A levels show a significantly increased fraction of cells with MN with 2.77% ± 0.48% (n = 3) and 1.95% ± 0.50% (n = 4) in CA$^{+/-}$ and CA$^{G/-}$ cells, respectively. Importantly, these two cell lines were derived independently from the parental CA$^{+/F}$ cell line (*Figure 1—figure supplement 1A*), which has wild-type levels of CENP-A and no significant increase in MN (*Figure 6*). In addition, neither cell line with increased CENP-A levels has a larger fraction of MN than CA$^{+/F}$ cells. While the essential role for CENP-A in centromere function is well established (*Régnier et al., 2005*; *Liu et al., 2006*; *Black et al., 2007*), our results indicate that a critical level of CENP-A is passed after reducing the levels to ~50%. However, the molecular mechanism responsible for MN formation remains unclear, as downstream centromere and kinetochore components of CENP-A remain unaffected.

## The contribution of cell type and local centromere features to centromeric CENP-A levels

Interestingly, we find that not all centromeres of the same cell have equal amounts of CENP-A (e.g., *Figure 5D*). This could either be due to *in cis* features driving differential regulation of CENP-A on individual centromeres, or by stochastic, yet unbiased, effects at centromeres. To distinguish between these possibilities, we measured the centromeric levels of endogenous CENP-A on specific chromosomes. First, we analyzed a monoclonal HCT-116 cell line that has an integrated Lac-array in a unique position in the genome (*Thompson and Compton, 2011*). While the site of integration is

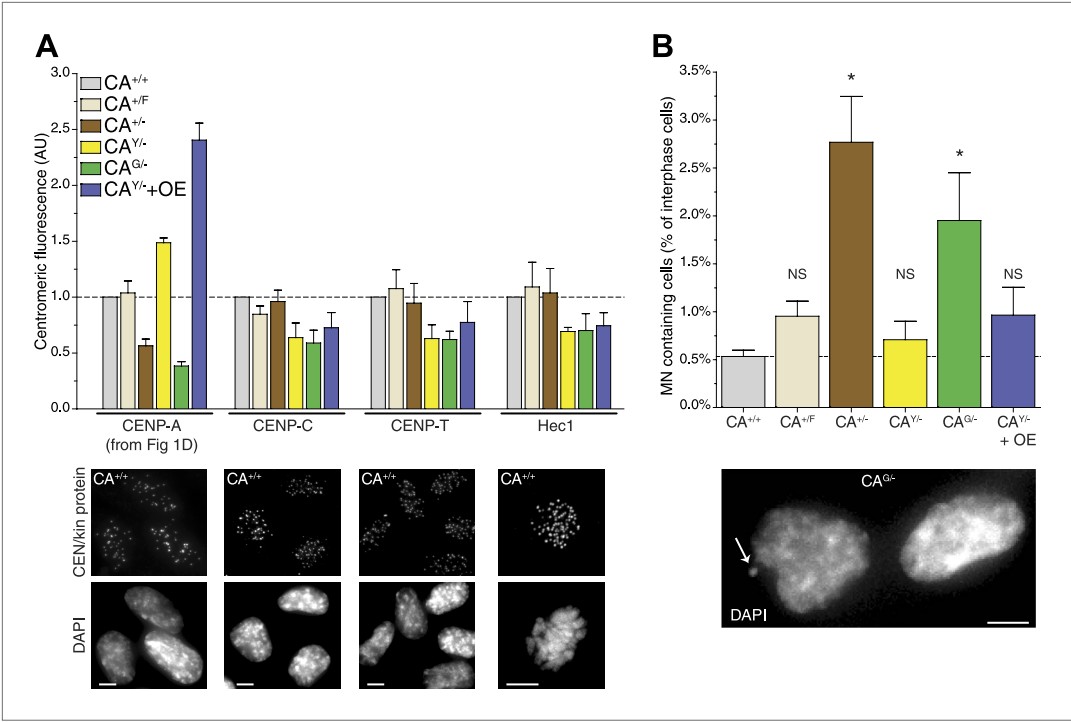

**Figure 6**. Reduction of CENP-A leads to a CENP-C, CENP-T, and Hec1 independent increase in micronuclei. (**A**) Quantification of centromeric CENP-A (from **Figure 1**), CENP-C, CENP-T, and Hec1 levels for indicated cell lines; n = 4 independent experiments in each case. Note that cell lines carrying tagged CENP-A have a slight, yet non-significant impairment in recruiting CENP-C, CENP-T, and Hec1. However, this does not correlate with the CENP-A levels themselves. Below, representative images of indicated antibody staining from CA$^{+/+}$ cells are shown. Representative images from all cell lines can be found in **Figure 6—figure supplement 1**. (**B**) Quantification of the fraction of cells containing micronuclei (MN) for indicated cell lines. Asterisk indicates statistically significant increase compared to wild-type (paired $t$ test; $p<0.05$; n = 3–4 independent experiments [500–3000 cells per experiment per cell line]); NS indicates no significant difference. Throughout, the average ± SEM is indicated and dashed lines represent wild-type levels. Scale bars: 5 µm.

The following figure supplements are available for figure 6:

**Figure supplement 1**. Representative images for quantifications in **Figure 6B**.

unknown, expressing LacI-GFP allows for the identification of the same chromosome in a population of cells (**Figure 7A**). Both the average and variance of CENP-A at this centromere does not differ statistically from the bulk (**Figure 7B**, **Figure 7—figure supplement 1A**), arguing against centromere specific features driving CENP-A levels on the Lac-marked chromosome. Conversely, we found that the Y-centromere, uniquely identified by the lack of CENP-B (**Figure 7C**; **Earnshaw et al., 1987**), of two independent male cell lines had a slight yet significant reduction of CENP-A (19% in wild-type HCT-116 and 13% in DLD-1; **Figure 7D**, **Figure 7—figure supplement 1B,C**), consistent with an earlier report (**Irvine et al., 2004**). Finally, we used a human patient-derived fibroblast cell line (PDNC-4) where one centromere of chromosome 4 has repositioned to an atypical location (**Amor et al., 2004**), which we designate as NeoCEN-4 (**Figure 7E**). As has been observed in other cell lines derived from this patient (**Amor et al., 2004**), we found that the NeoCEN-4 has a ~25% decrease in centromeric CENP-A (**Figure 7F**, **Figure 7—figure supplement 1D**). Taken together, these results show that while CENP-A expression drives centromeric levels, local sequence or chromatin features can also contribute to the average amount of CENP-A at specific centromeres. Nevertheless, even on these centromeres, the variance in CENP-A levels is maintained, indicating that other stochastic processes contribute to CENP-A levels.

Next, to determine whether the CENP-A copy number of our model cell line is representative for functionally different cells, we performed comparative immunofluorescence against CENP-A

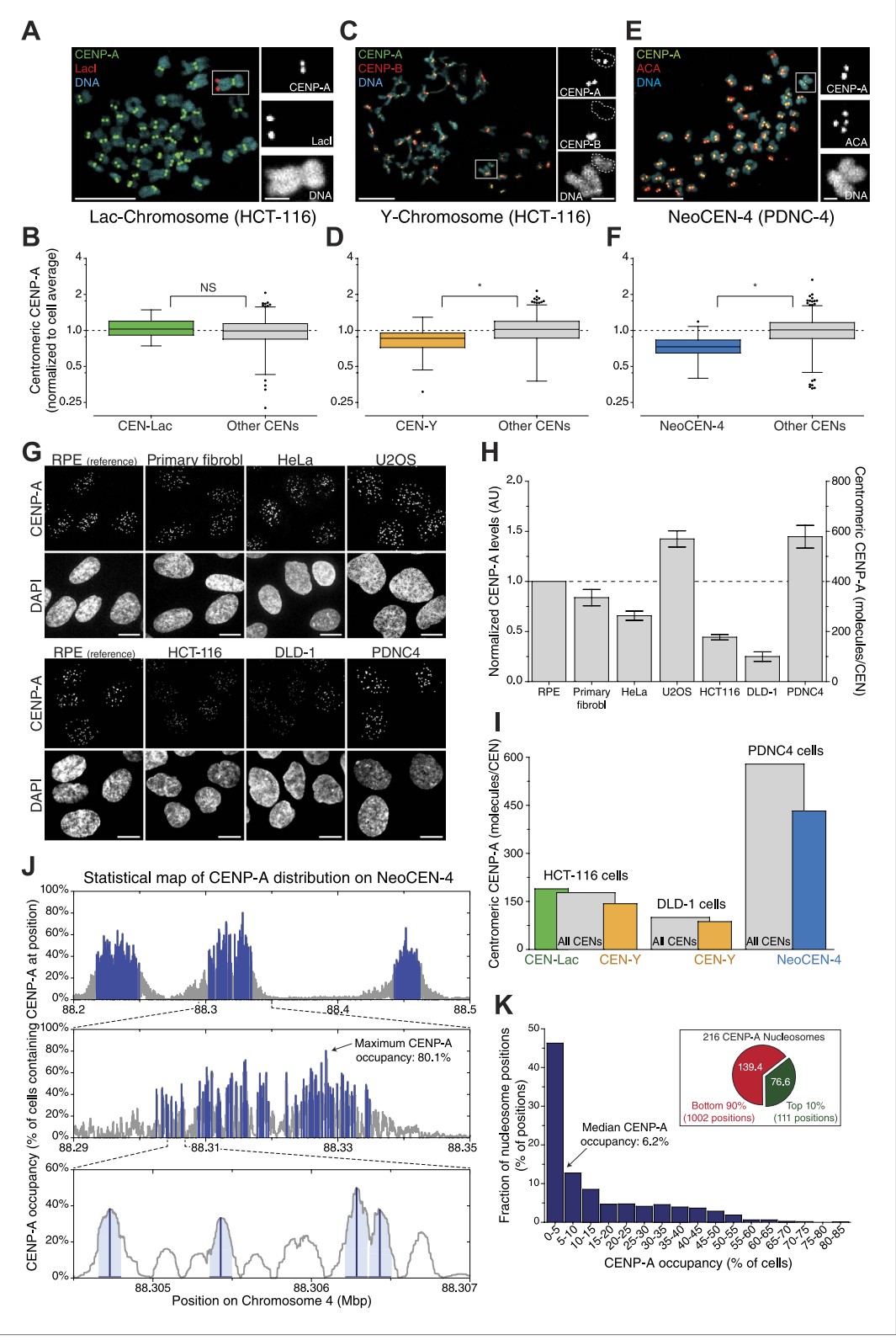

**Figure 7**. Centromere and cell specific distribution of CENP-A. (**A**, **C**, **E**) Representative micrograph of mitotic spreads for LacI-GFP::LacO expressing HCT-116 cells (**A**); wild-type HCT-116 cells (**C**); and PDNC-4 cells (**E**). Blowups show the chromosome containing the integrated Lac-array (**A**); the Y-chromosome (outline indicated; CENP-B negative) as well as an autosome (CENP-B positive) (**C**); and the neocentric chromosome 4, containing

*Figure 7. Continued on next page*

*Figure 7. Continued*

2 pairs of ACA spots (staining both CENP-A and CENP-B), but only 1 pair of CENP-A spots (**E**). (**B**, **D**, **F**) Quantification of CENP-A levels on the centromere of the chromosome containing the Lac-array (CEN-Lac; n = 29; **B**); the Y-chromosome (CEN-Y; n = 18; **D**); and neocentric chromosome 4 (NeoCEN-4; n = 39; **F**) of indicated cell lines compared to all other centromeres within the same cell (Other CENs; n = 1008, 620, and 1592, respectively). Median (line), interquartile distance (box), 3 × interquartile distance or extremes (whiskers), and outliers (spots) are indicated. *Figure 7—figure supplement 1* shows results of individual centromeres. Asterisk indicates statistically significant difference (*t* test; p<0.05); NS indicates no significant difference. (**G**) Representative images of CENP-A antibody staining in indicated cell types. Images of RPE cells are shown as independent reference. Primary fibrobl. indicates primary human foreskin fibroblasts. (**H**) Quantification of **G**. Mean ± SEM for n = 3–4 independent experiments is shown. Left y-axis represents centromeric CENP-A levels normalized to RPE cells; right y-axis shows number of CENP-A molecules per centromere (CEN). (**I**) Combined results from **A**–**H** allow for the determination of CENP-A copy numbers on individual chromosomes as indicated. (**J**) Statistical map of the distribution of 216 CENP-A nucleosomes on the NeoCEN-4 at three different scales. The top 216 peaks are indicated in blue. Y-axis indicates the probability of CENP-A occupancy for each nucleosome. (**K**) Histogram of the CENP-A nucleosome occupancy. Inset shows the distribution of 216 neocentric CENP-A nucleosomes on the 10% highest occupancy peaks (green) and 90% lowest occupancy peaks (red).

The following figure supplements are available for figure 7:

**Figure supplement 1**. Measurements of individual centromeres for graphs in *Figure 7A–F*.

---

(*Figure 7G*). We analyzed four different cancer cell lines (HeLa, U2OS, HCT-116, and DLD-1), as well as the PDNC-4 neocentromere cell line discussed above, and primary human foreskin fibroblasts that were cultured for a limited number of passages (<15) since their isolation from a patient (*Figure 7G*). Using these cell lines, we found a sixfold range of centromeric CENP-A levels (*Figure 7H*), indicating that there is substantial variance between different cell lines. However, we find that the primary cells have a similar amount of CENP-A as RPEs (*Figure 7H*), arguing that our measure of absolute CENP-A copy numbers made in RPE cells is relevant for healthy, human tissues as well.

We combined these results with our measurements of individual centromeres and determined that, while an average centromere in PDNC-4 cells contains ~579 molecules of CENP-A, the NeoCEN-4 only contains ~432. Average Y-centromeres contain ~143 or ~87 molecules in HCT-116 and DLD-1 cells, respectively (*Figure 7I*). In conclusion, we find evidence that *cis*-elements can have an effect on CENP-A levels, at least on human Y- and neo-centromeres.

## A statistical map of CENP-A occupancy at individual nucleosome positions

The number of CENP-A nucleosomes we find at individual centromeres is much smaller (~25-fold, see *Figure 8A*) than the total number of nucleosome positions on human centromeric DNA. This indicates that either CENP-A is randomly distributed at a low level throughout the centromere domain or that it occupies specific 'hotspots'. Due to their repetitive nature, it is not possible to map individual CENP-A nucleosomes on canonical centromeres. However, a recent high-resolution ChIP-seq analysis of the (non-repetitive) NeoCEN-4 identified 1113 unique CENP-A nucleosome positions spanning a ~300 kb locus (*Hasson et al., 2013*). By combining the relative height of individual peaks with the total number of CENP-A nucleosomes at this neocentromere, we were able to determine the fraction of cells containing CENP-A at each nucleosome position (*Figure 7J*). This statistical map of CENP-A occupancy shows that, while the median is ~6% (*Figure 7K*), individual positions feature CENP-A with a surprisingly high occupancy (up to 80% of all cells; *Figure 7J*, arrow). Remarkably, more than one third of all CENP-A nucleosomes are located on the top 10% potential positions (*Figure 7K*, inset). This strongly suggests that, at least on the NeoCEN-4, a number of nucleosome positioning sequences exist that strongly favor CENP-A over other H3 variants.

## Discussion

It has been proposed that centromeres in budding yeast feature a single nucleosome of CENP-A$^{Cse4}$ (*Meluh et al., 1998*; *Furuyama and Biggins, 2007*). For this reason, the yeast centromere cluster has been extensively used to calibrate fluorescence intensities of CENP-A and other proteins from a

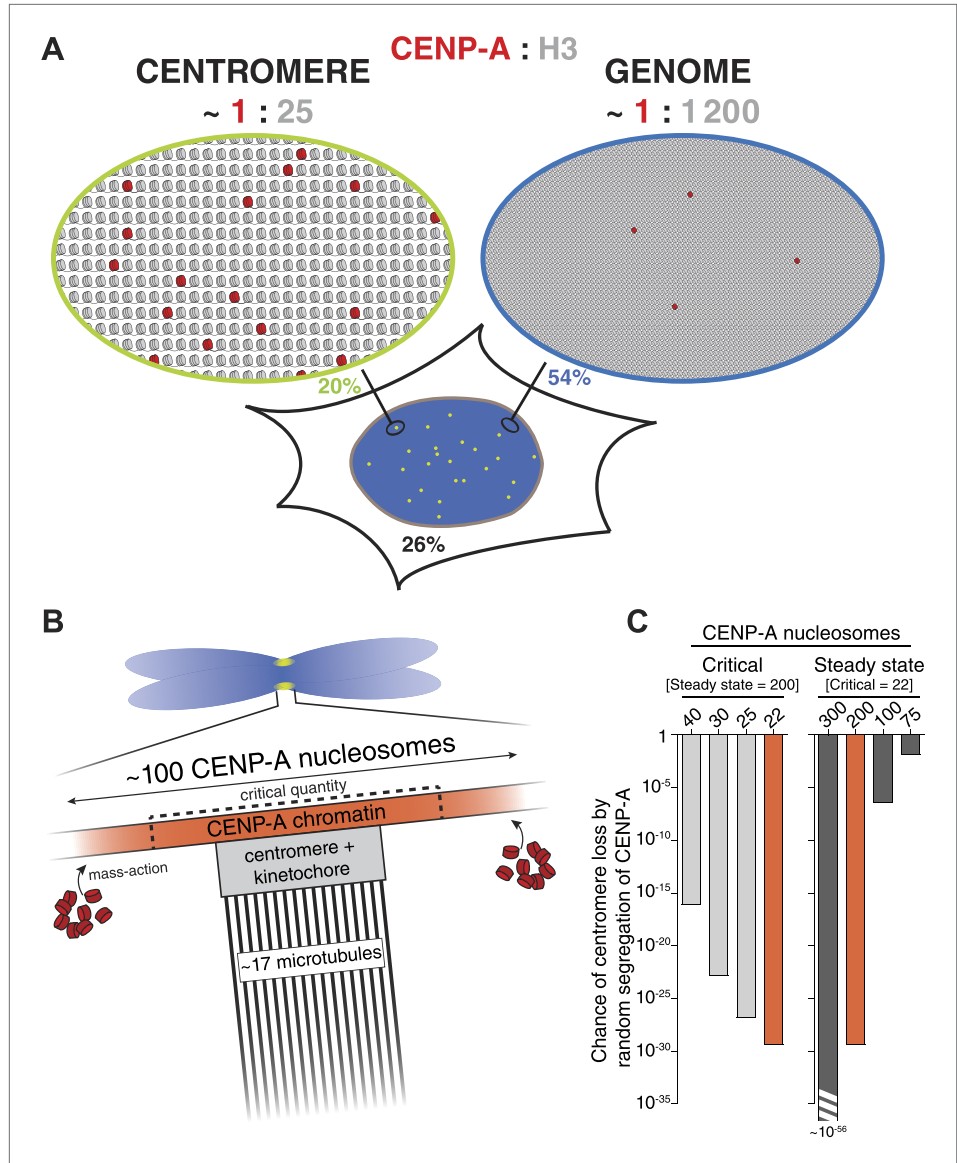

**Figure 8**. A quantitative view of human centromeric chromatin. (**A**) Distribution of CENP-A. Estimated ratio of CENP-A (red) to H3 (gray) at the centromere and on non-centromeric loci (genome) in interphase cells. Estimations are calculated assuming 2 CENP-A molecules per nucleosome (*Sekulic et al., 2010*; *Tachiwana et al., 2011*; *Bassett et al., 2012*; *Hasson et al., 2013*; *Padeganeh et al., 2013*), an average nucleosome positioning distance of 200 base pairs, an average centromere size of $2.5 \times 10^6$ base pairs (*Sullivan et al., 1996*; *Lee et al., 1997*) of which approximately 40% (1 Mbp) contains CENP-A (*Sullivan et al., 2011*), a diploid genome size of $6 \times 10^9$ base pairs, 200 CENP-A nucleosomes per centromere, and $2.5 \times 10^4$ CENP-A nucleosomes outside of centromeres ($9.1 \times 10^4$ CENP-A molecules per cell [*Figure 2F*], of which 74% is in chromatin [*Figure 4B*] and 0.44% in each centromere [*Figure 2B*]). The fraction of CENP-A on centromeres, non-centromeric chromatin, and unincorporated CENP-A are indicated in green, blue, and black, respectively. CENP-A nucleosomes are represented as though evenly spread throughout the centromeric domain. Alternatively, they could be distributed into one or more clusters within this domain. (**B**) Mitotic organization of centromeric chromatin. 200 nucleosomes are redistributed to 100 nucleosomes per centromere on replicated sister chromatids (*Jansen et al., 2007*; *Bodor et al., 2013*). The exact CENP-A copy number at the centromere depends on the available total pool (mass-action). Excess CENP-A could either lead to an increased CENP-A domain or lead to a higher density of CENP-A within a domain of fixed size. This pool forms an excess to recruit downstream centromere and kinetochore complexes and ultimately provides microtubule binding sites for ~17 kinetochore microtubules (*McEwen et al., 2001*). To avoid mitotic errors, a critical amount of CENP-A is required (dashed lines). (**C**) Graph representing the chance of at least one

*Figure 8. Continued on next page*

*Figure 8. Continued*

chromosome in a cell (with 46 chromosomes) reaching critically low levels of CENP-A by random segregation of pre-existing CENP-A nucleosomes. Calculations were performed for varying levels of critical nucleosome numbers at a fixed steady state of 200 (left), or by varying the steady state number at a fixed critical level of 22 (right). Red bars represent identical calculations.

number of species (*Joglekar et al., 2006*, *2008*; *Johnston et al., 2010*; *Schittenhelm et al., 2010*). However, the single nucleosome hypothesis has recently been challenged (*Coffman et al., 2011*; *Lawrimore et al., 2011*; *Haase et al., 2013*). To avoid dependency on any single reference, we used three independent methods to measure the human centromeric CENP-A copy number. One strategy uses intrinsically controlled fluorescence ratios of cellular and centromeric CENP-A-YFP signals (*Figure 2A*). The second method does not rely directly on fluorescence intensities, but rather on the stochastic redistribution of CENP-A (*Figure 5A*). Finally, we compared CENP-A signals directly to a calibrated fluorescent standard (*Figure 5E*). Importantly, despite the independent nature of these strategies, they all come to a very similar conclusion. Therefore, we demonstrate that typical centromeres in human RPE cells contain ~400 molecules of CENP-A. While there is a continuing debate on the composition of CENP-A nucleosomes (*Black and Cleveland, 2011*; *Henikoff and Furuyama, 2012*), current evidence, at least in human cells, strongly favors an octameric arrangement harboring two copies of CENP-A (*Sekulic et al., 2010*; *Tachiwana et al., 2011*; *Bassett et al., 2012*; *Hasson et al., 2013*; *Padeganeh et al., 2013*). Hence, our numbers, correspond to 200 CENP-A nucleosomes in interphase, which are split into 100 nucleosomes on mitotic chromosomes (*Figure 8B*).

Epigenetic centromere inheritance is achieved by quantitative inheritance of CENP-A across cell division cycles (*Jansen et al., 2007*; *Bodor et al., 2013*). We find that rather than accurately ensuring that each daughter receives exactly half, redistribution of CENP-A occurs in a random fashion (*Figure 5B,C*). Because this type of regulation has the potential for individual centromeres to stochastically inherit critically low levels of CENP-A, the steady state must be sufficiently high to avoid chromosome loss. Although the critical amount of CENP-A is not known, we have previously shown that HeLa cell viability is lost if CENP-A levels are reduced to ~33% (*Black et al., 2007*), *i.e.* 44 nucleosomes (see *Figure 7H*). Conversely, we show here that CA$^{G/-}$ cells are viable at 40% of RPE levels (80 nucleosomes). Consequently, we estimate that the critical number of nucleosomes that must be inherited, which is half of the steady state level and is replenished during G1 phase, lies between 22 and 40. We used these values to calculate the chance that any one centromere per cell inherits critically low levels of CENP-A for different steady state and critical CENP-A nucleosome levels (*Figure 8C*). We demonstrate that at a steady state of 200 CENP-A nucleosomes per centromere, less than one in $10^{16}$ cell divisions will give rise to a centromere containing 40 CENP-A nucleosomes or less (*Figure 8C*, left). Thus, the chance of inheriting a critical amount of CENP-A at wild-type steady state levels is negligible. Conversely, with 100 CENP-A nucleosomes at steady state, the chance of a chromosome inheriting even the most stringent critical level of 22 nucleosomes is close to $10^{-6}$ (*Figure 8C*, right), which may pose a significant problem, for example during the development of a human organism. Conversely, although critical levels would be reached even less frequently if centromeres contained a steady state of, for example 300 CENP-A nucleosomes, this degree of accuracy may be superfluous and not outweigh the cost of maintaining a large centromere size (*Figure 8C*, right). Therefore, we argue that the number of CENP-A molecules found on human centromeres is optimized for robust epigenetic inheritance and centromeric function.

Previously, it has been shown that CENP-A is interspersed with both H3.1 and H3.3 at the centromere (*Blower et al., 2002*; *Sullivan and Karpen, 2004*; *Ribeiro et al., 2010*; *Dunleavy et al., 2011*; *Sullivan et al., 2011*). Indeed, based on the average size of the centromeric chromatin domain, we estimate that 200 CENP-A nucleosomes represent only ~4% of all centromeric nucleosomes (see *Figure 8A* for calculation). Surprisingly, we find that the majority of chromatin bound CENP-A is located outside the centromere. Indeed, a recent study found that a proportion of CENP-A containing nucleosomes also exist in non-centromeric chromatin of HeLa cells, and is assembled by DAXX, a major chaperone of histone H3.3 (*Lacoste et al., 2014*). In addition, detectable levels of non-centromeric CENP-A have been observed in budding yeast (*Camahort et al., 2009*) and chicken DT40 cells (*Shang et al., 2013*). Here, we quantify this pool in human RPE cells and while there is more than twice

as many non-centromeric CENP-A nucleosomes than there are centromeric ones, this only represents <0.1% of all nucleosomes in the genome and thus CENP-A is ~50-fold enriched (per unit length of DNA) at centromeres (*Figure 8A*). This result may explain how, despite being outnumbered 25:1 by other H3 variants at the centromere, CENP-A can still accurately specify the centromeric locus. This hypothesis may potentially be tested by creating artificial CENP-A binding sites (e.g., using the LacO/LacI system) of different known sizes and determining the threshold at which centromeres can be formed.

Interestingly, the study by Lacoste et al. showed that the extra-centromeric CENP-A is not randomly distributed, but enriched at sites of high histone turnover (*Lacoste et al., 2014*). Our finding that CENP-T, CENP-C, and Hec1 do not quantitatively correlate with CENP-A levels (*Figure 6A*) argues that not each (non-centromeric) CENP-A nucleosome is able to recruit downstream centromere components. It would be interesting to determine to what extent other centromere and kinetochore proteins are present throughout the genome and whether they are also enriched at extra-centromeric CENP-A 'hotspots'. This question is particularly relevant since it has been observed that downstream centromere components may affect centromeric CENP-A levels (*Okada et al., 2006*; *Carroll et al., 2009*, *2010*; *Hori et al., 2013*). A critical combination of components at such hotspots may trigger neocentreomere formation, the mechanisms of which are still unresolved.

Previously, it has been observed that at very high levels of overexpression, CENP-A ceases to be centromere restricted (*Van Hooser et al., 2001*; *Heun et al., 2006*; *Gascoigne et al., 2011*). Nevertheless, here we show that within a sixfold range of expression levels, the CENP-A loading machinery has a constant efficiency, which maintains a strict ratio between the centromeric and total pools of CENP-A. Thus, within a physiological range, centromeric CENP-A levels are regulated by a mass-action mechanism, where the loading efficiency is independent of the expression levels. This mechanism ensures that with fluctuating expression levels, CENP-A remains mainly centromere restricted, and may prevent potential neocentromere seeding.

Remarkably, varying the amount of CENP-A at centromeres during perpetual growth in culture does not affect the levels of several other centromeric proteins. One possible explanation for this is that there is a fixed subset of 'active' CENP-A nucleosomes that is responsible for recruiting downstream components, even if the total amount of CENP-A is variable. Alternatively, an excess of CENP-A could form a chromatin domain that provides a 'platform' for recruitment of a centromere complex of fixed size. Surprisingly, however, we find that a critical amount of CENP-A for prevention of micronuclei is reached even before downstream centromere and kinetochore protein levels are affected (*Figures 6 and 8B*).

Our analysis indicates that the distribution of CENP-A among centromeres within one cell is generally uniform. However, in agreement with prior publications, we find that both the Y-centromere and a human neocentromere have lower CENP-A levels (*Amor et al., 2004*; *Irvine et al., 2004*). Interestingly, both these centromere types are atypical in that they are formed on relatively small genomic loci: ~600 kb for the Y-centromere (*Abruzzo et al., 1996*) and ~300 kb for the NeoCEN-4 (*Hasson et al., 2013*), whereas autosomes and the X-chromosome have alpha-sattelite arrays of several magabases in size (*Wevrick and Willard, 1989*; *Mahtani and Willard, 1990*; *Lo et al., 1999*). In addition, in contrast to canonical centromeres, neither the Y-centromere nor neocentromeres recruit the sequence-specific DNA binding protein CENP-B (*Earnshaw et al., 1987*; *Amor et al., 2004*), which has been hypothesized to alter the 3D structure of centromeric chromatin (*Pluta et al., 1992*). The presence of CENP-B binding sites has recently been shown to have a role in phasing CENP-A nucleosomes (*Hasson et al., 2013*), and to cooperate with CENP-A in kinetochore function (*Fachinetti et al., 2013*), and may therefore be involved in regulation of centromeric CENP-A levels as well. Furthermore, high resolution analysis of a human neocentromere reveals a non-random distribution of CENP-A (*Hasson et al., 2013*), where individual nucleosome positions are occupied in anywhere between 0.5% and 80% of cells (*Figure 7J,K*). Thus, despite specific DNA sequences being neither sufficient nor required for centromere identity (*Earnshaw and Migeon, 1985*; *Voullaire et al., 1993*; *Amor et al., 2004*; *Marshall et al., 2008*), the amount of CENP-A at centromeres likely results from a combination of a systematic cellular mechanism with a contribution of local sequence or chromatin features.

In conclusion, several key mechanistic insights follow from our findings. First, while CENP-A nucleosomes are highly enriched at the centromere, most CENP-A is distributed at low levels throughout chromatin. This indicates that there is no exclusive pathway that restricts CENP-A assembly to centromeres. Nevertheless, we propose that the ample number of CENP-A nucleosomes facilitates a

robust epigenetic signal that can absorb fluctuations in CENP-A inheritance and assembly in order to faithfully maintain centromere identity. Secondly, the requirement for a sizable number of CENP-A nucleosomes to perpetuate an active centromere reduces the likelihood for inadvertent detrimental neocentromere seeding without the need for a tightly restricted assembly mechanism. In addition, the fixed ratio between total and centromeric CENP-A levels may prevent excess CENP-A from accumulating at high density at non-centromeric loci, thus further reducing the probability of neocentromere formation. Finally, the number of centromeric CENP-A nucleosomes represents an ample pool of which only a subset is required to nucleate otherwise self-organized centromere and kinetochore complexes. In summary, from our analysis an integrated view of centromeric architecture, size, and regulation emerges (*Figure 8*) that provides a basis to explain the self-propagating nature of the epigenetic centromere.

## Materials and methods

### Cell culture and construction

All human cell lines used were grown at 37°C, 5% $CO_2$. Cells were grown in DMEM/F-12 (RPE), DMEM (HeLa, U2OS, PDNC-4), MEM (primary fibroblasts; Coriell GM06170), McCoy's 5A (HCT-116), or RPMI-1640 (DLD-1) cell culture media. Media were supplemented with 10% fetal bovine serum (FBS), 2 mM glutamine, 1 mM sodium pyruvate (SP), 100 U/ml penicillin, and 100 µg/ml streptomycin, with the following exceptions: for RPE cells SP was substituted for 14.5 mM sodium bicarbonate; for HeLa newborn calf serum was used instead of FBS; for fibroblasts 15% FBS was used; for DLD-1 cells SP was omitted; and both SP and glutamine were omitted for HCT-116 cells. During live cell imaging, culture medium was replaced with Leibowitz's L-15 medium containing 10% FBS and 2 mM glutamine. LacI-GFP::LacO HCT-116 cells (gift from Duane Compton, *Thompson and Compton, 2011*) were selected alternatingly with 2 µg/ml blasticidin and 300 µg/ml hygromycin; PDNC-4 cells were selected with 100 µg/ml neomycin. All media and supplements were purchased from Gibco (Paisley, UK).

All targeted cell lines are derived from wild-type hTERT RPE cells ($CA^{+/+}$). Gene targeting was achieved by Adeno-associated virus (AAV) mediated delivery of targeting constructs essentially as described (*Berdougo et al., 2009*), except in the case if $CA^{G/-}$ cells (see below). The $CA^{+/F}$ cell line was created by inserting loxP sites surrounding CENP-A exons 2 and 4 as described previously (*Fachinetti et al., 2013*). The $CA^{+/-}$ cell line was created by targeting the floxed CENP-A allele of $CA^{+/F}$ cells with a construct lacking 1373 bp of the CENP-A gene (from 43 bp upstream of exon 2 to 134 bp downstream of exon 4) encompassing the essential CENP-A targeting domain (*Black et al., 2007*). $CA^{Y/-}$ cells were created by sequential targeting of a first CENP-A allele with the targeting construct inserting loxP sites flanking exon 3 and 4 as described above and the second allele by targeting EYFP (carrying citrine and monomerization mutations: Q69M, A206K) in frame with the CENP-A gene, immediately prior to the stop codon in exon 4. The floxed allele was subsequently removed by retroviral delivery of HR-MMPCreGFP, a 'Hit and Run' Cre vector, as described (*Silver and Livingston, 2001*). $CA^{G/-}$ cells were created from an independent $CA^{+/-}$ clone where the remaining intact CENP-A allele was targeted with EGFP using a FACS-based strategy that we developed previously (*Mata et al., 2012*). Targeting resulted in insertion of the EGFP ORF directly downstream the last coding sequence in exon 4, just upstream of the endogenous stop codon, without insertion of any selectable marker gene. $CA^{Y/-}$+OE cells were created by stable transfection of and selection (5 µg/ml blasticidin) for a CENP-A-YFP expression vector (pBOS-Blast) bearing a CENP-A-YFP fusion protein identical to that of the endogenous knockin locus in $CA^{Y/-}$ cells. $CA^{Y/-}$+H2B-RFP and $CA^{+/+}$+H2B-RFP cell lines were created by stable transfection of and selection (5 µg/ml puromycin) for a H2B-RFP expression vector (*Black et al., 2007*) in $CA^{Y/-}$ and $CA^{+/+}$ cells, respectively. All cell lines were monoclonally sorted by FACS.

For the transient transfection experiment (*Figure 1F*), wild-type HeLa cells were first synchronized in S phase by addition of 2 mM thymidine. After 17 hr, cells were released using 24 µM deoxycytidine and simultaneously transfected with untagged, wild-type CENP-A and/or HJURP expression vectors (or an empty vector) in combination with an EYFP-CENP-C expression vector (*Shah et al., 2004*) (2:2:1 proportion). 9 hr later, thymidine was re-added for an additional 15 hr, at which point cells were again released with deoxycytidine for 9 hr. A final thymidine arrest was performed and after 15 hr cells were fixed. Only cells expressing the positive transfection marker EYFP-CENP-C were analyzed. All stable and transient transfections were performed using Lipofectamine LTX (Invitrogen; Carlsbad, CA) according to the manufacturer's instructions.

## Immunoblotting and cell fractionation

All samples were prepared in 1X Laemmli sample buffer, separated by SDS-PAGE, and transferred onto nitrocellulose membranes. Whole cell extracts were prepared by lysing cells directly in sample buffer, to ensure that the entire cellular protein pool remained present in the sample. Recombinant CENP-A/H4-complexes were purified as described previously (*Black et al., 2004*), concentration was determined by $A_{280}$ measurement and mixed with protein extracts from chicken DT40 cells to bring the overall protein concentration of the purified CENP-A protein preps to a level comparable to the RPE cell extracts. Absence of cross-recognition of human CENP-A antibody to chicken protein was confirmed by omission of recombinant human CENP-A protein in DT40 extracts (*Figure 2D*, second lane). Alternatively, recombinant CENP-A/H4 was spiked into RPE cell extracts. Results obtained from the two methods are comparable (95.3 ± 14.0 ng [n = 8] and 75.4 ± 5.4 ng [n = 2], respectively; p>0.5). Cellular CENP-A quantity was determined by comparison of fluorescence derived from cellular and purified CENP-A. The following antibodies and dilutions were used: CENP-A (#2186; Cell Signaling Technology, Danvers, MA or *Ando et al., 2002*) at 1:1000 or tissue culture supernatant at 1:400, respectively; α-tubulin (DM1A; Sigma-Aldrich, St. Louis, MO) at 1:5000; HJURP (gift from Dan Foltz, *Foltz et al., 2009*) at 1:10,000; Mis18BP1 (A302-825A; Bethyl Laboratories, Inc., Montgomery, TX) at 1:2000; H4K20me2 (ab9052; Abcam, Cambridge, UK) at 1:1000. IRDye800CW-coupled anti-mouse or anti-rabbit (Licor Biosciences) and DyLight680-coupled anti-mouse or anti-rabbit (Rockland Immunochemicals, Gilbertsville, PA) secondary antibodies were used prior to detection on an Odyssey near-infrared scanner (Licor Biosciences, Lincoln, NE). Immunoblot signals were quantified using the Odyssey software, and a linear response was confirmed over a 32-fold range (*Figure 2E*). Target protein signals were normalized to the α-tubulin loading control signal to correct for slight deviations in cell concentration between extracts of different RPE cell lines.

Cell fractionation was performed for CA$^{Y/-}$+H2B-RFP cells after cell lysis in ice cold buffer consisting of 50 mM Tris-HCl (pH 7.5), 150 mM NaCl, 0.5 mM EDTA, 1% Triton-X 100, 1 mM DTT, and a mix of protease inhibitors (1 mM PMSF, 1 µg/ml leupeptin, 1 µg/ml pepstatin, and aprotinin [A6279; Sigma, 1:1000 dilution]). Soluble proteins were separated from the insoluble fraction by centrifugation at 21,000×g at 4°C and resuspended in an equal volume of lysis buffer. Both supernatant and pellet fractions were incubated with 1.25 U/µl of benzonase nuclease (Novagen, San Diego, CA) on ice for 30 min prior to denaturation in Laemmli sample buffer.

## Microscopy

Imaging was performed on an Andor Revolution XD system, controlling an inverted microscope (Eclipse-Ti; Nikon, Tokyo, Japan), an iXonEM+ EMCCD camera (DU-897; Andor, Belfast, UK), a CSU-X1 spinning disk unit (Yokogawa, Tokyo, Japan), a laser combiner/multi-port switch system (Andor) and a motorized stage (Prior Scientific, Cambridge, UK), controlled by MicroManager software (*Edelstein et al., 2010*). Images were collected using a Nikon 100X, 1.4 NA, Plan Apo oil immersion objective (fixed cell imaging) or a Nikon 60X, 1.2 NA, Plan Apo VC water immersion objective (live cell imaging) at 1× binning. For live cell imaging, the temperature of the chamber was maintained at 37°C.

## Fluorescence lifetime measurements

Cells grown on glass coverslips were fixed and mounted as described (*Bodor et al., 2012*) and imaged using a Zeiss LSM710 coupled to a motorized stage of an upright Zeiss Axio Examiner microscope equipped with a Zeiss 63X, 1.4 NA, Plan Apo oil immersion objective lens. A Coherent Chameleon Vision II multi-photon Ti-Sapphire laser was used to excite EYFP samples. All images were 512 × 512 pixels in size, with a pixel size of 0.09 µm. For all samples, an optimal setting of the laser power and PMT voltage was chosen to avoid pixel saturation and minimize photobleaching. The CLSM settings were kept constant so that valid comparisons could be made between measurements from different samples. Fluorescence lifetime imaging microscopy (FLIM) was performed by measuring the decay rate of EYFP using a Becker & Hickl time-correlated single photon counting hybrid detector coupled to the confocal LSM710 setup. The SPCImage (Becker & Hickl, Berlin, Germany) software was utilized to perform single exponential fitting for each pixel location.

## Immunofluorescence and mitotic spreads

Cell fixation, immunofluorescence, and DAPI staining was performed as described previously (*Bodor et al., 2012*). The following antibodies and dilutions were used: CENP-A (gift from Tatsuo Fukagawa,

*Ando et al., 2002*) tissue culture supernatant at 1:100, rabbit polyclonal CENP-B (sc22788; Santa Cruz Biotechnology, Dallas, TX) at 1:100, tissue culture supernatant from mouse hybridomas expressing monoclonal CENP-B (*Earnshaw et al., 1987*) at 1:4, CENP-C (*Foltz et al., 2009*) at 1:10,000, CENP-T (gift from Dan Foltz, *Barnhart et al., 2011*) at 1:1000, Hec1 (9G3.23; MA1-23308; Pierce, Rockford, IL) at 1:100, ACA (anti-centromere antibodies; 83JD, gift from Kevin Sullivan) at 1:100. Fluorescent secondary antibodies were obtained from Jackson ImmunoResearch (West Grove, PA) or Rockland ImmunoChemicals and used at a dilution of 1:200. Immunofluorescence signals of *Figures 1C, 5E, 6B, 7G* were automatically quantified using the CRaQ method as described previously (*Bodor et al., 2012*) using CENP-T or CENP-C as a centromere reference. Hec1 levels were measured exclusively in prometaphase or metaphase (based on DAPI staining) of unperturbed cells. Micronuclei were scored based on DAPI staining.

Mitotic spreads were performed after mitotic shake-off of cells arrested overnight (~16 hr) in 250 ng/ml nocodazole. 25,000 cells/ml were swollen in 75 mM KCl and 5000 cells were cytospun onto coverslips using a Cytopro 7620 cytocentrifuge (Wescor Inc., Logan, UT) for 4 min, at 1200 rpm, high acceleration. Cells were then fixed and processed for immunofluorescence as described above. Average centromere signals of both sisters were measured after background correction, by subtracting the minimum pixel value from the maximum of a box of 5 × 5 pixels around each sister centromere. Specific chromosomal markers were used as described in the text to detect centromeres of interest and signals were normalized to the average of all centromeres of the same cell spread.

## Quantification of the centromeric CENP-A copy number

CA$^{+/+}$ cells were mixed with CA$^{Y/-}$, CA$^{G/-}$, or CA$^{Y/-}$+OE cells at a ~1:4 ratio on 35 mm glass-bottom petri dishes (MatTek Corporation, Ashland, MA). Non-cell permeable dextran-AlexaFluor647 (10,000 MW; Molecular Probes, Eugene, OR) was added at 2–4 µg/ml to stain the medium outside of cells (*Figure 2A,I*). To minimize oversampling, individual live cells were imaged at 500 nm axial resolution (close to the resolution limit of the objective) spanning the entire cell volume. Images were flatfield corrected for unequal illumination using the signal of a uniform fluorescent slide and the 'Shading Corrector' plugin for FIJI. For each axial section, the cell outline was determined based on absence of dextran-AlexaFluor647 staining, and the integrated fluorescence intensities inside the cell outline as well as those of 1–3 independent background regions per section were determined. Background corrected signals from all sections were summed to determine the total cellular fluorescence. Fluorescence measurements of CA$^{Y/-}$, CA$^{G/-}$, or CA$^{Y/-}$+OE cells were corrected for autofluorescence by subtraction of average per pixel fluorescence intensity of non-fluorescent CA$^{+/+}$ cells from the same dish. Alternatively, CA$^{+/+}$+H2B-RFP and CA$^{Y/-}$+H2B-RFP cells were mixed and no dextran was added to the medium. In this case, the H2B-RFP signal was used to determine the nuclear volume, and the total nuclear fluorescence was determined as described above for the total cellular volume. Automated centromere detection was performed by an analogous algorithm to a previous study (*Bodor et al., 2012*, *2013*), where diffraction limited spots are detected based on their size, circularity, and feret's diameter. Centromere signals were measured by integrating the intensity of a 5 pixel diameter surrounding each centromere in the appropriate axial section. Local background fluorescence was derived by measuring the difference in intensity between concentric circles of 5 and 7 pixel diameter, and subtracted from centromeric signals (*Hoffman et al., 2001*). In addition, centromeric signals were corrected for axial oversampling. For this, diffraction limited spots of yellow/green PS-Speck fluorescent beads (Molecular Probes) were measured in multiple plains. The sum intensity of individual beads from all these plains was compared to the signal in the plain with the maximum signal (*i.e.*, the focal plane). The percentage of centromeric fluorescence was determined in relationship to the total fluorescence of each individual cell.

To allow for cell cycle staging of CA$^{Y/-}$ cells, transduction with hCdt1(30/120)-RFP was performed using the BacMam 2.0 baculovirus system (Invitrogen). Expression levels of transduced protein were allowed to stabilize for 3 days prior to analysis. Individual cells were followed by live cell microscopy using DIC and RFP signals. Nuclear RFP signals were tracked every ~2 hr to monitor their cell cycle progression. Imaging of CENP-A-YFP and cellular volume were performed as described above. Analysis of the centromeric CENP-A ratio was performed as described above, but restricted to cells in which RFP levels were decreasing at the specific timepoint of analysis (to exclude cells in G1 phase) and which did not enter mitosis or showed an increase in RFP levels for at least the following 3–4 hr (to exclude cells in G2 phase). Centromeric ratio was compared to non-transduced, randomly cycling cells (*Figure 3C*) or randomly cycling cells that were transduced, but not followed over time

(*Figure 3—figure supplement 1*). For these experiments, wild-type cells used to measure cellular autofluorescence were seeded on a separate dish.

## Stochastic fluctuation measurements

$CA^{Y/-}$, $CA^{G/-}$ or $CA^{Y/-}$+OE cells were treated with nocodazole (250–500 ng/µl) for ~9 hr, after which cells were fixed and processed for immunofluorescence as described above. Sister centromere pairs were identified by CENP-B staining and GFP or YFP fluorescence intensity of each sister was measured and background corrected by subtracting the minimum pixel value of a 5 pixel diameter circle from the maximum value. The difference ($\delta$) in fluorescence intensity and the sum ($\Sigma$) intensity of the two sisters were determined. The fluorescence intensity per segregating unit ($\alpha$) was determined from the average $\delta^2/\Sigma$ of all centromere pairs of the same experiment and cell line. The number of segregating units on each centromere was calculated as $\Sigma/\alpha$, as described previously (*Rosenfeld et al., 2005*, *2006*) and in *Figure 5A*. In addition to sister centromeres, three independent rounds of random centromere pairing between all centromeres measured in a single experiment on $CA^{G/-}$ cells were performed and centromeric CENP-A-GFP units based on these pairings were quantified in *Figure 5—figure supplement 1E*.

## Yeast growth and imaging

4 kb-LacO, LacI-GFP *Saccharomyces cerevisiae* (gift from Kerry Bloom, *Lawrimore et al., 2011*) were grown in minimal synthetic media (Yeast nitrogen base [Sigma] + complete synthetic defined single drop-out medium lacking uracil and histidine [MP Biomedicals, Solon, OH]), supplemented with 2% D(+) Glucose (Merck, Darmstadt, Germany). Prior to imaging, log-phase cells ($OD_{600}$ of ~0.7) were transferred onto a 2% low melting agarose pad and sealed under a coverslip with VALAP (1:1:1 vaseline:lanolin:paraffin). $CA^{G/-}$ cells were grown on 35-mm glass-bottom petri dishes and yeast and human cells were imaged using identical settings during the same microscopy session. Fluorescence intensity of centromeres and Lac-arrays were quantified after background correction (maximum minus minimum of a 5 × 5 pixel box).

## Integrating ChIP-seq and quantitative data of CENP-A at a human neocentromere

CENP-A ChIP-Seq data from the PDNC-4 neocentromere cell line (Accession #GSE44724) was processed as previously described (*Hasson et al., 2013*). Briefly, paired-end ChIP-Seq reads were aligned to the human genome build hg19 with Bowtie2 version 2.0.0 using paired-end mode. Reads were aligned by using a seed length of 50 bp, and only the single best alignment per read with up to two mismatches was reported in the SAM file. The aligned mate pairs were joined in MATLAB by requiring ≥95% overlap identity. The joined reads were aligned to the PDNC-4 neocentromere and only reads which mapped with 100% identity were used in the subsequent analysis. Nucleosome positions at the neocentromere were determined using the '*findpeaks*' function in MATLAB. The probability of CENP-A occupancy at a given position was determined according to the following formula: (total reads overlying that position) × (216 CENP-A nucleosomes [*Figure 7I*])/(total reads mapping to the entire neocentromere).

## Calculation of the chance of reaching critical CENP-A levels after random segregation

All calculations represented in *Figure 8C* were performed in R. For these calculations we assume that CENP-A is inherited following a binominal distribution, consistent with our findings (*Figure 5*, *Figure 5—figure supplement 1A,C*). To determine the chance (X) of any chromosome reaching critical levels of CENP-A, the '*pbinom*' function was used to calculate the fraction of a binomial distribution (where p=0.5 and n [steady state number of nucleosomes] = 200 or was varied as indicated) that is either below a critical value (c = 22, or varied as indicated) or above a critical value (n−c). To determine the chance that any chromosome in a cell (containing 46 chromosomes) reaches critical levels, we calculated the chance that 46 independent centromeres do not reach critical levels and subtracted this chance from 1; $[1 - (1 - X)^{46}]$.

## Acknowledgements

We thank Tatsuo Fukagawa (National Institute of Genetics, Shizuoka, Japan), Dan Foltz (University of Virginia, Charlottesville, VA), Kevin Sullivan (National University of Ireland, Galway, Ireland), David Livingston (Dana-Farber Cancer Institute, Boston, MA), Bernardo Orr, and Duane Compton (Dartmouth Medical School, Hanover, NH), and Kerry Bloom (University of North Carolina, Chapel Hill, NC) for reagents, Nitzan Rosenfeld (Cancer Research UK, Cambridge, UK) for advice, and Jorge Carneiro

(Instituto Gulbenkian de Ciência, Oeiras, Portugal) for help using R. We thank the Confocal and Light Microscopy core facility at Dana Farber Cancer Institute (Harvard Medical School) for providing access to the FLIM setup. We are grateful to Alekos Athanasiadis and Monica Bettencourt-Dias (both at Instituto Gulbenkian de Ciência, Oeiras, Portugal) for helpful comments on the manuscript.

## Additional information

### Funding

| Funder | Grant reference number | Author |
| --- | --- | --- |
| European Molecular Biology Organization | EMBO Installation grant | Lars ET Jansen |
| European Commission | FP7 Marie Curie Reintegration grant | Lars ET Jansen |
| European Research Council | ERC-2013-CoG-615638 | Lars ET Jansen |
| National Institutes of Health | GM082989 | Ben E Black |
| National Institutes of Health | GM077238 | Jagesh V Shah |
| Burroughs Wellcome Fund | | Ben E Black |
| Rita Allen Foundation | | Ben E Black |
| Beckman Laser Institute and Foundation | | Jagesh V Shah |
| Fundação para a Ciência e a Tecnologia (Foundation for Science and Technology) | SFRH/BD/74284/2010 | Dani L Bodor |
| Fundação para a Ciência e a Tecnologia (Foundation for Science and Technology) | BIA-BCM/100557/2008, BIA-PRO/100537/2008 | Lars ET Jansen |

The funders had no role in study design, data collection and interpretation, or the decision to submit the work for publication.

### Author contributions

DLB, Conception and design, Acquisition of data, Analysis and interpretation of data, Drafting or revising the article; JFM, Contributed to construction and characterization of knockout/knockin cell lines; MS, Acquisition of fluorescence life time data; AFD, Performed acute CENP-A/HJURP induction experiment; KJS, TP, Analysis of sequencing data; DWC, Contributed to the establishment of CENP-A floxed and knockout alleles; BEB, Analysis of sequencing data, Contributions to concept and design; JVS, Analysis of fluorescence life time data, Contributions to concept and design; LETJ, Conception and design, Drafting or revising the article

## Additional files

### Major dataset

The following previously published dataset was used:

| Author(s) | Year | Dataset title | Dataset ID and/or URL | Database, license, and accessibility information |
| --- | --- | --- | --- | --- |
| Hasson D, Panchenko T, Salimian KJ, Salman MU, Sekulic N, Alonso A, Warburton PE, Black BE | 2013 | Genome-wide maps of CENP-A nucleosomes in three neocentromere-containing cell lines | GSE44724; http://www.ncbi.nlm.nih.gov/geo/query/acc.cgi?acc=GSE44724 | Publicly available at NCBI Gene Expression Omnibus. |

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
