## [Decision Letter]

[Editors’ note: although it is not typical of the review process at *eLife*, in this case the editors decided to include the reviews in their entirety for the authors’ consideration as they prepared their revised submission.]

Thank you for sending your work entitled “The quantitative architecture of centromeric chromatin” for consideration at *eLife*. Your article has been favorably evaluated by a Senior editor, a Reviewing editor, and 3 reviewers.

The Reviewing editor and the other reviewers discussed their comments before we reached this decision, and the Reviewing editor has assembled the following comments to help you prepare a revised submission.

Two primary concerns were raised.

1) Whether the quantitative immunoblotting is accurate. Reviewer 2 recommends quantitative blots of the discarded insoluble material to demonstrate that all CENP-A has been extracted. Then to demonstrate that CENP-A is not passing through the filters blotting should be done with a stack of two or (better) three filters, and demonstrating that CENP-A cannot be detected on the second and third filter.

2) Reviewer 3 suggests that averaging the quantitation over an asynchronous population leads to an inaccurate estimate of CENP-A loading. This may explain the variation between individual cells mentioned by reviewer 1. To control for this the reviewers suggest selecting cells at G1/S to represent cells that completed centromeric CENP-A assembly (100% centromere occupancy and minimal soluble pool) and late G2 or mitotic cells to represent cells with maximal nascent soluble pool, then perform the quantitation on these isolates. These populations should be obtained without using drug synchronisation that could perturb the process, for example by mitotic shake off. In this respect, Reviewer 3 emphasises (point 5) that your argument that low levels of CENP-A could account for mis-segregation would not stand if sisters with less CENP-A are 'replenished' during the subsequent G1.

These points will require more experimental data; the other points below should be addressed by re-writing the manuscript notably those concerning potential over-interpretation of the data. None of the referees was persuaded by your conclusions that the generation of micronuclei is a surrogate marker for centromere dysfunction.

Reviewer 1 Comments:

1) It would have been interesting to complement 1E by showing a similar graph for a protein that does not follow mass-action (e.g. a kinetochore component?).

2) Why do GFP and YFP constructs yield considerably different results with the stochastic fluctuation method?

3) The authors assume that a large portion of CENP-A is assembled into non-centromeric chromatin. Is there any evidence that these CENP-A molecules are indeed incorporated into nucleosomes/chromatin (rather than being chromatin-associated in a less defined manner)?

4) 2B/3A [currently Figures 2 and 4]: In my opinion the average should have been calculated by averaging the mean of single cells rather than averaging over all centromeres (which gives some cells more weight than others).

5) The y error bars in 1E do not match the error bars in 1D, although the data should be identical.

6) I would have appreciated yet more detail on the experimental methods. For example: (a) although described in the previous paper, briefly mention by which criteria centromeres are included in or excluded from the quantification; (b) how is the value for axial oversampling determined?; (c) I do not understand the rational for subtracting the minimum pixel value, since this is already background-subtracted (Immunofluorescence and mitotic spreads section); (d) comment on differences/similarities between mixing recombinant CENP-A/H4 into chicken or human cell extract.

7) The text covering Figure 1B/C could be improved. Make clear that immunostaining can be compared between cell lines. In contrast, I would not state that GFP and YFP blotting can be compared.

8) 5B [currently Figure 6] suggests that the tagged versions are not fully functional, which could be mentioned.

9) 7B [currently Figure 8]erroneously implies that abundance correlates with spatial expansion - should be removed or revised. 7C [currently Figure 8]: labeling could be improved to make the figure easier to understand.

Reviewer 2 Comments:

1) Measuring the cellular level of a protein by immunoblotting is tricky. Here a bit more work is needed by the authors. They need to provide some sort of convincing control that they are measuring all of the cellular CENP-A.

Firstly, we need quantitative blots of the discarded insoluble material to demonstrate that all CNP-A has been extracted. Secondly, they need to demonstrate that CENP-A is not passing through the filters they are using for quantitative immunoblotting. This is done by doing the blots with a stack of two or (better) three filters, and demonstrating that CENP-A cannot be detected on the second and third filter. When an analogous experiment was done with CENP-B, the protein was detected even unto the third filter. Despite this concern, I am reasonably content with the conclusion that the average centromere contains ∼400 molecules.

2) In my opinion, the authors state, but do not fully explore their most important conclusion - e.g. that “about one-fifth of the CENP-A protein content (0.44% x 46) is present on the functionally relevant subcellular location”. The authors should say “centromeres” instead of “the functionally relevant subcellular location”.

3) The authors state that neocentromeres are “fully functional centromeres that have repositioned to atypical loci on the chromosome”. Can the authors cite a study that has actually demonstrated that neocentromeres are fully functional (e.g. a study that has measured their efficiency at directing chromosome segregation in a quantitative manner)?

4) I am less convinced by the attempts of the authors at assessing the critical number of CENP-A nucleosomes required for centromere function. Clearly budding yeast can do this with ∼1 nucleosome, and Bruce Nicklas showed years ago that a single microtubule could move a huge newt chromosome in the cytoplasm. Furthermore, the authors' failure to see alterations in levels of CNP-C, -T or Ndc80 when the minimum “critical” threshold level of CENP-A is passed suggests that the explanations for changes in the micronucleus index that they see when CENP-A levels drop below 50%.

5) I was interested by the statement that “Interestingly, we find that not all centromeres of the same cell have equal amounts of CENP-A”. The authors should carry out a similar quantitation on CENP-C and CENP-T.

6) The fact that the Y centromere has less CENP-A seems to impress the authors, but they seem to ignore an obvious possible explanation for this: the Y chromosome also has by far the smallest centromeric alpha-satellite DNA array. In my experience, neocentromeres also have lower levels of core centromere proteins (these authors also look at this and confirm this conclusion). Thus, I believe that in their speculations on the explanation for this phenomenon, the authors should include the overall size of the alphoid array in addition to specific sequence preferences.

7) The authors state that “cis-elements can have an effect on CENP-A levels, at least on atypical human Y- and neocentromeres”. Is the Y centromere atypical? Every male human has one.

8) To me the authors have chosen to make light of the most amazing result of this paper - that 80% of the CENP-A on chromosomes is not at centromeres! Others have found that there are significant amounts of CENP-A outside centromeres, but have not tried to quantify it as convincingly as these authors. For example in Camahort et al. (Mol Cell v. 35), the fact that significant amounts of Cse4 were found outside the centromere was mentioned. Recently in the neocentromere paper by Hori et al. (Dev. Cell), their ChIP-Seq data revealed significant amounts of CENP-A outside the chicken centromeres Z and 5, but again overall amounts were not given. The authors should expand their discussion to explore this. For example, how much of a non-centromeric pool is there likely to be for other CCAN components? The answer could be similar to that for CENP-A. Thus, the key question is what causes the critical local concentration that nucleates the centromeric chromatin?

9) It would not detract from the import of the present work if the authors here mentioned the fact that their numbers are not terribly dissimilar from those estimated by Riberio et al. (PNAS), particularly given that the latter were looking at a different organism.

10) The statement “Previously, it was shown that CENP-A is interspersed with both H3.1 and H3.3 at the centromere (11; 21).” should also reference the Ribeiro et al. study.

11) The authors go on in the same paragraph to say “Nevertheless, this non-centromeric pool only represents <0.1% of all nucleosomes in the genome and thus CENP-A is ∼50-fold enriched (per unit length of DNA) at centromeres (Figure 7a [currently Figure 8]). This explains how, despite being outnumbered 25:1 by other H3 variants at the centromere, CENP-A can still accurately specify the centromeric locus.” Are they claiming to offer a convincing explanation of how the CENP-A accurately specifies the centromere? If they really believe this, perhaps they could expand and come up with a more directly testable model.

12) Where the authors say “Indeed, it has recently been shown that this sequence specific DNA binding protein has a role in phasing CENP-A nucleosomes (36)”, they might like to read, and possibly cite, the rather extensive discussion of this in Pluta et al. (JCB 116, 1081, pp. 1091, 1092). The discussion of the enigmatic CENP-B aside, I feel (as already stated above) that the reason for the low levels of CENP-A on the Y chromosome most likely arises from the small size of the alphoid array, which the authors should mention here as well.

13) The authors state “the fixed ratio between total and centromeric CENP-A levels prevents excess CENP-A from accumulating at high density at non-centromeric loci”. I believe that this is an overstatement of what the data actually show. This is still, in my mind, a hypothesis - not a rigorously proven fact, and should be stated as such.

14) Lastly, I do not understand Figure 7b [current Figure 8]. What is the “critical size” and why is this bigger than the kinetochore?

Reviewer 3 Comments:

The authors should discuss possible limitations of their findings, and be careful when making concluding statements. For example, the authors tried to correlate further reduction of CENP-A from the parental RPE line with increases in micronuclei. I am not convinced that the slightly increased frequency of micronuclei is due to a moderate decrease in CENP-A. They would need to show that restoring CENP-A suppresses the phenotype. It is especially a problem when the authors attempt to say that this amount of CENP-A represents the number in “normal” tissue. The results are meaningful in the background of the RPE cell line, but even it is a relevant number, it still has limited value since they have shown that CENP-A levels vary greatly among cell lines and there is no strict correlation with the level of genome instability. This suggests that other 'background' mutations probably contribute greatly to the level of genome instability, and that they only made a weak correlation between CENP-A reduction and increased micronuclei. Overall, these issues raise concerns about using micronucleus frequencies as a direct measure of the impact of altering CENP-A levels, and whether CENP-A reduction accounts for the observed phenotype. The authors determined the critical number based on the observations that HeLa cells with 33% CENP-A are not viable but RPE cells with 40% CENP-A are viable, so they reason that the critical number must lie somewhere in between these two numbers. However, these are completely different lines, with HeLa cells carrying hundreds or thousands of other mutations not in the RPE background. Viability is a combined readout of more than one related pathway. Viability of HeLa cells and RPE cells may have different sensitivity to the levels of CENP-A due to potential mutations in related pathways.

There are several places the authors cite improper literatures, please cite original studies. For example, in introduction, to make the point that CENP-A is so far the primary candidate for centromere identity, the authors need to cite papers beyond just studies on human CENP-A; in the next sentence, to show the point that CENP-A is stably transmitted through cell divisions, there are much more authoritative papers on humans and other organisms than the authors' own papers; the paper cited for primary DNA sequence being neither necessary nor sufficient for centromere determination is Amor et al. in 2004, which is not the best for making this point.

In several places the authors stated results as if they are original observations. However, some of the principles have been revealed or suggested previously, and are here reinforced by the measurements in this manuscript. For example, the correlation between reduction of CENP-A and increased genome instability (if indeed due to CENP-A reduction in the RPE background), and different amounts of CENP-A are at different centromeres. The authors should be careful and need to cite the literature properly.

1) Local background correction for Figure 2 is problematic. The basis of doing local background correction is the assumption that signals immediately surrounding centromeres are noise. What is the factual foundation of this methodology considering that recent papers and the authors' own data indicated the existence of non-centromeric CENP-A in chromatin? If the local background is indeed extremely low, treating it as background noise may not distort the actual number significantly.

2) Figure 1, to overcome the difficulty of different transfer efficiency for untagged or tagged CENP-A, the authors could also use known quantities of purified tagged and untagged CENP-A on the same gel to calibrate tagged CENP-A cell lines.

3) What is the cause of the observed different levels of CENP-A for different constructs in Figure 1? Did the authors compare different clones? This may suggest that other mutations occurred during generation of these lines along with changes in CENP-A levels.

4) Figure 1f, this experiment needs to be performed in the RPE lines not in HeLa. In addition, since transient CENP-A overexpression leads to incorporation into non-centromeric chromatin through a DAXX dependent pathway, this must represent a re-equilibrium of two different loading systems that utilize different assembly factors (HJURP and DAXX), and between centromeres and non-centromeric loci. Does DAXX mediated incorporation happen in the various RPE cells? The authors only considered canonical CENP-A loading at the centromere, but I don't think that it makes much sense to talk about mass action without considering the DAXX dependent assembly that may compete with HJURP for centromeric assembly. This will clearly influence the size of the available CENP-A pool for centromeric loading.

5) Figure 4b and 4c [current Figure 5], the authors measured fluorescence associated with mitotic centromeres. They found that different centromeres behave differently in the same cell. Using data from multiple cells, they concluded that segregation of CENP-A nucleosomes is random. To conclude this, the authors should have compared the behavior of an individual pair between different cells. It is possible that segregation is non-random for a given centromere of the same chromosome, but varies between centromeres of different chromosomes. Combining data from all centromeres without separating individual chromosomes may give rise to the frequency graph in Figure 4c [current Figure 5]. In addition, the authors need to consider that any differences in the levels of CENP-A at different centromeres/sisters observed in mitosis could be counteracted during new CENP-A assembly in the subsequent G1 phase.

6) The variation of numbers of CENP-A per centromere using different cell lines is over 2.5 fold, which is fairly large. What could be contributing to this variation?

7) Figure 5 [current Figure 6], the micronuclei phenotype has very limited value in assessing centromere defects. It is actually problematic as the end phenotype for centromere dysfunction, since there could be defects in sister separation (cohesion), as well as genome instability/breakage. It is more direct and informative to look at chromosome segregation. Without additional experiments, e.g. live studies, the results and their implications are hard to interpret. The authors also need to evaluate more than one clone for each line.

8) For the phenotype in Figure 5 [current Figure 6], how is it striking to see a correlation between reduced CENP-A and increased genome instability? More important, it is not known whether this is due to the problem of the tags or other potential mutations arising during generation of the cell lines. To conclude that the micronuclei phenotype is caused by reduction of CENP-A, the authors would have to do a rescue experiment to restore CENP-A to normal.

9) Figure 6 [current Figure 7], the authors made a statement about the variance of CENP-A between untransformed cells and cancer cells based on few lines from each category. This statement should be removed. It is safe to say that variance in cancer cell lines is big (6-fold range), but I don't think the evidence is sufficient to say anything about untransformed cells.

10) In the Discussion, please rephrase “the fixed ratio between total and centromeric CENP-A levels prevents excess CENP-A from accumulating at high density at non-centromeric loci, thus further reducing the probability of neocentromere formation.” Fixed ratio is an observation it should not be taken as a mechanism to explain how neo-centromeres are prevented.

11) Also the model figure shows widely spaced single CENPA nucleosomes and could easily be clustered more.

---

## [Author Response]

*1) Whether the quantitative immunoblotting is accurate. Reviewer 2 recommends quantitative blots of the discarded insoluble material to demonstrate that all CENP-A has been extracted. Then to demonstrate that CENP-A is not passing through the filters blotting should be done with a stack of two or (better) three filters, and demonstrating that CENP-A cannot be detected on the second and third filter*.

In response to the first point raised, regarding discarded insoluble material, we would like to emphasize that to ensure no protein was lost during preparation of the lysates, whole cell extracts were prepared without clearance of the insoluble protein pool in all of our experiments (except for the fractionation experiments in Figure 4). We now clarified this in the main text, in Figure 2, and in the Methods section.

In addition, we performed the experiment suggested by the reviewer, where we blot CENP-A onto a stack of three membranes (Figure 2—figure supplement 3). The reviewer was correct that CENP-A can indeed transfer through the membranes. However, importantly, we show that the amount of CENP-A retained on the first membrane (which is quantified throughout) is the same between recombinant and cellular (untagged) CENP-A. Therefore, we remain confident that a quantitative comparison between CENP-A derived from these two sources is accurate. In addition, we found that tagged CENP-A does not pass through the first filter, which reinforces our statement that we cannot quantitatively compare proteins of different sizes by immunoblotting.

*2) Reviewer 3 suggests that averaging the quantitation over an asynchronous population leads to an inaccurate estimate of CENP-A loading. This may explain the variation between individual cells mentioned by reviewer 1. To control for this the reviewers suggest selecting cells at G1/S to represent cells that completed centromeric CENP-A assembly (100% centromere occupancy and minimal soluble pool) and late G2 or mitotic cells to represent cells with maximal nascent soluble pool, then perform the quantitation on these isolates. These populations should be obtained without using drug synchronisation that could perturb the process, for example by mitotic shake off. In this respect, Reviewer 3 emphasises (point 5) that your argument that low levels of CENP-A could account for mis-segregation would not stand if sisters with less CENP-A are 'replenished' during the subsequent G1*.

The reviewer is correct that one possible source of variation between individual cells may be cell cycle related. However, performing the suggested experiment is challenging at best. We found that cells rapidly lose synchrony after mitotic shake off (even before a majority of cells reaches S phase). Instead, we address this issue in a manner that avoids using drug-based synchronization using part of the FUCCI system developed by Sakaue-Sawano et al. (Cell, 2008). With this system, we were able to selectively measure the centromeric CENP-A ratio in S phase cells. Because in human cells CENP-A expression starts in (late) G2 phase (Shelby et al., JCB 2000) and incorporation occurs in (early) G1 phase (44), by selecting cells in S phase, we likely discard any variation of centromeric to whole cell fluorescence that would be induced by cell cycle related effects.

We expressed hCdt1(30/120)-RFP in CA^Y/-^ cells using a commercially available baculovirus system. This fluorescent marker is expressed in all stages of the cell cycle, but degraded during S and G2 phase and mitosis. Thus, RFP fluorescence of cells entering into and progressing through S phase decreases, while signal increases again in G1 phase. To select for S phase cells, we followed cells live by microscopy and screened the RFP signal. We selectively analyzed CENP-A fluorescence in cells in which the RFP signal decreased with respect to previous time points (thus excluding cells in G1 phase), and which did not enter mitosis and/or started increasing RFP fluorescence for the following 4 hours after analysis (thus excluding G2 and mitotic cells).

In summary, we found that in S phase, neither the mean centromeric ratio nor the variance differs from randomly cycling cells. The description and results of this experiment have been added to our manuscript in the main text and Methods and in the current Figure 3 and Figure 3—figure supplement 1.

Reviewer 3 states that our argument that low levels of CENP-A could account for mis-segregation would not stand if sisters with less CENP-A are 'replenished' during the subsequent G1. We are aware of such proposals where centromeres with critically low levels would selectively recruit more CENP-A (e.g. Brown and Xu, BioEssays 2009). However, to our knowledge there is no experimental evidence for such a mechanism. In addition, when CENP-A drops below critical levels e.g. during S phase, this may lead to mitotic failure in the same cell cycle, even before replenishment which happens later in G1 phase. Therefore, our argument that centromeres with critically low CENP-A levels are likely resulting in centromere failure is not unreasonable and with the present state of knowledge the most parsimonious.

*These points will require more experimental data; the other points below should be addressed by re-writing the manuscript notably those concerning potential over-interpretation of the data. None of the referees was persuaded by your conclusions that the generation of micronuclei is a surrogate marker for centromere dysfunction*.

We have now extensively rewritten the section of our manuscript that concerns the determination of critical CENP-A levels to more accurately reflect what the results show. In brief, we have attenuated our conclusions regarding the increased number of micronuclei present in cells with lowered CENP-A levels. We removed the suggestion that this is indicative of mitotic errors and do not make any conclusive statement regarding the molecular nature of micronucleus formation in these cells. In addition, we have now put more emphasis on the fact that downstream centromere/kinetochore protein recruitment does not quantitatively rely on the amount of CENP-A present. Importantly, this last result argues against an existing model of a modular centromere architecture that is assembled from repeating structural substructures, initially proposed by Zinkowski et al. in 1991.

*Reviewer 1*
*Comments:*

*1) It would have been interesting to complement 1E by showing a similar graph for a protein that does not follow mass-action (e.g. a kinetochore component?)*.

We agree that this is an interesting issue. However, we feel that it lies outside of the scope of our study.

*2) Why do GFP and YFP constructs yield considerably different results with*
*the stochastic fluctuation method?*

The stochastic fluctuation method measures the number of what we designated as ‘segregating units,’ i.e. the number of fluorophores that co-segregate. There is extensive (albeit indirect) evidence that existing octameric nucleosomes containing two CENP-A molecules do not split in two during DNA replication. Therefore, a minimum of two molecules of CENP-A co-segregate, thus forming a segregating unit. However, the extent of co-segregation may also depend on the density of CENP-A nucleosomes in the centromere, which would lead to the difference we observed between our cell lines (i.e. CENP-A-YFP being expressed at higher levels). Thus, without making assumptions on the exact nature of the co-segregating unit, we cite measurements from the stochastic fluctuation method as a ‘minimal estimate’ of the CENP-A copy number throughout the manuscript. Importantly, independent of the cell line used, we find that results obtained with the stochastic fluctuation method agree within an approximately 2-fold range with the results from the integrated fluorescence approach (Figure 2). We have modified the section on this quantification method to help clarify the issue raised above.

*3) The authors assume that a large portion of CENP-A is assembled into non-centromeric chromatin. Is there any evidence that these CENP-A molecules are indeed incorporated into nucleosomes/chromatin (rather than being chromatin-associated*
*in a less defined manner)?*

Although nucleosome incorporation is likely the case (in part taking into account a study in Molecular Cell by the Almouzni lab that appeared in print during submission of our manuscript to *eLife),* it is true that we have no direct evidence for the status of chromatin bound CENP-A. We now refer to this pool CENP-A as “chromatin bound”.

*4) 2B/3A [currently *Figures 2 and 4*]: In my opinion the average should have been calculated by averaging the mean of single cells rather than averaging over all centromeres (which gives some cells more weight than others)*.

We realize that there are valid arguments for either way of averaging centromeric fractions. However, in our particular datasets the different methods for averaging give near-identical results. Specifically, averaging as suggested by the reviewer would result in the following values (number between brackets shows averages over all centromere, as performed in the original submission): CA^Y/-^: 0.43% (0.44%), CA^G/-^: 0.37% (0.38%), CA^Y/-^+OE: 0.38% (0.38%), and CA^Y/-^+H2B-RFP: 0.77% (0.73%).

*5) The y error bars in 1E do not match the error bars in 1D, although the data should be identical*.

We thank the reviewer for pointing out this error in the figure. We had inadvertently used the standard deviation in Figure 1 rather than the standard error of the mean, which is the measure used in Figure 1 and throughout the manuscript. We have now corrected this in Figure 1 and Figure 1—figure supplement 1.

*6) I would have appreciated yet more detail on the experimental methods. For example: (a) although described in the previous paper, briefly mention by which criteria centromeres are included in or excluded from the quantification; (b) how is the value for axial oversampling determined?; (c) on page 25, I do not understand the rational for subtracting the minimum pixel value, since this is already background-subtracted (Immunofluorescence and mitotic spread section); (d) comment on differences/similarities between mixing recombinant CENP-A/H4 into chicken or human cell extract*.

We have added experimental details as suggested for points (a), (b), and (d). Regarding point (c), we have now clarified in the text that subtraction of the minimum pixel value is in fact the background correction method applied.

*7) The text covering*
Figure 1*/C could be improved. Make clear that immunostaining can be compared between cell lines. In contrast, I would not state that GFP and YFP blotting can be compared*.

The first point is now emphasized in the legend of Figure 1.

Regarding the second point: despite the few (8) amino acid differences (and small apparent size difference) between CENP-A-YFP and CENP-A-GFP, we feel that these fusion proteins are still directly comparable by Western Blot and have no indications that this would be problematic. This is in sharp contrast to the comparison between proteins of vastly different sizes such as untagged CENP-A (140 amino acids; ∼16 kDa) and tagged CENP-A-GFP or -YFP (385 amino acids, ∼43 kDa). See also the new Figure 2—figure supplement 3 regarding this issue.

*8) 5B suggests that the tagged versions are not fully functional, which could be mentioned*.

This is now mentioned in the figure legend (current Figure 6)

*9) 7B erroneously implies that abundance correlates with spatial expansion - should be removed or revised. 7C: labeling could be improved to make the figure easier to understand*.

Current Figure 8: The figure legend was adapted to include the alternative possibility of having an increased density of CENP-A within a domain of fixed size.

Current Figure 8: The y-axis was relabeled for clarity and further explanation was added to the legend.

*Reviewer 2*
*Comments:*

*1) Measuring the cellular level of a protein by immunoblotting is tricky. Here a bit more work is needed by the authors. They need to provide some sort of convincing control that they are measuring all of the cellular CENP-A*.

*Firstly, we need quantitative blots of the discarded insoluble material to demonstrate that all CNP-A has been extracted. Secondly, they need to demonstrate that CENP-A is not passing through the filters they are using for quantitative immunoblotting. This is done by doing the blots with a stack of two or (better) three filters, and demonstrating that CENP-A cannot be detected on the second and third filter. When an analogous experiment was done with CENP-B, the protein was detected even unto the third filter. Despite this concern, I am reasonably content with the conclusion that the average centromere contains ∼400 molecules*.

As this issue has been raised as one of the primary concerns regarding our manuscript, we addressed it above.

*2) In my opinion, the authors state, but do not fully explore their most important conclusion - e.g. that “about one-fifth of the CENP-A protein content (0.44% x 46) is present on the functionally relevant subcellular location”. The authors should say “centromeres” instead of “the functionally relevant subcellular location”*.

We agree with the reviewer that our finding that only a minority of CENP-A is centromere localized is an important one. We have now emphasized this point in the text. We have also emphasized that by the functionally relevant location we do in fact mean at the centromere.

*3) The authors state that neocentromeres are “fully functional centromeres that have repositioned to atypical loci on the chromosome”. Can the authors cite a study that has actually demonstrated that neocentromeres are fully functional (e.g. a study that has measured their efficiency at directing chromosome segregation in a*
*quantitative manner)?*

We have removed the word ‘fully’, as it has in fact been shown that at least one particular neocentromere is less efficient at error correction than endogenous centromeres (Bassett et al., JCB 2010)

*4) I am less convinced by the attempts of the authors at assessing the critical number of CENP-A nucleosomes required for centromere function. Clearly budding yeast can do this with ∼1 nucleosome, and Bruce Nicklas showed years ago that a single microtubule could move a huge newt chromosome in the cytoplasm. Furthermore, the authors' failure to see alterations in levels of CNP-C, -T or Ndc80 when the minimum “critical” threshold level of CENP-A is passed suggests that the explanations for changes in the micronucleus index that they see when CENP-A levels drop below 50%*.

Unfortunately, the last part of this comment was lost and it is not completely clear to us what the reviewer meant. Nevertheless, we accept the criticism that the increase in micronuclei at low CENP-A levels does not necessarily result from centromere failure. As outlined above, we have rewritten this section of the manuscript to more accurately represent the observations made.

*5) I was interested by the statement that “Interestingly, we find that not all centromeres of the same cell have equal amounts of CENP-A”. The authors should carry out a similar quantitation on CENP-C and CENP-T*.

We show in Figure 6 that CENP-C and -T levels do not correspond directly the levels of CENP-A. Furthermore, we provide evidence in Figure 7 that a large proportion of the variation seen in the levels of CENP-A is stochastic in nature. Thus, while the CENP-C and -T levels are likely also variable between centromeres, it is unlikely that this variation would be caused by variable CENP-A levels. Based on these arguments we feel that the rather extensive analysis proposed by the reviewer lies beyond the scope of our study.

*6) The fact that the Y centromere has less CENP-A seems to impress the authors, but they seem to ignore an obvious possible explanation for this: the Y chromosome also has by far the smallest centromeric alpha-satellite DNA array. In my experience, neocentromeres also have lower levels of core centromere proteins (these authors also look at this and confirm this conclusion). Thus, I believe that in their speculations on the explanation for this phenomenon, the authors should include the overall size of the alphoid array in addition to specific sequence preferences*.

The reviewer is correct that the size of Y centromere and neocentromere is indeed another difference between these centromeres and other centromeres and could be causative for the different levels of CENP-A. We have added this hypothesis to the Discussion, regarding why these centromeres carry less CENP-A.

*7) The authors state that “cis-elements can have an effect on CENP-A levels, at least on atypical human Y- and neocentromeres”. Is the Y centromere atypical? Every male human has one*.

We have removed the term atypical from this section of the paper. We explain in what sense we consider these two types of centromeres atypical, namely that they lack CENP-B and are formed on relatively small genomic loci.

*8) To me the authors have chosen to make light of the most amazing result of this paper - that 80% of the CENP-A on chromosomes is not at centromeres! Others have found that there are significant amounts of CENP-A outside centromeres, but have not tried to quantify it as convincingly as these authors. For example in Camahort et al. (Mol Cell v. 35), the fact that significant amounts of Cse4 were found outside the centromere was mentioned. Recently in the neocentromere paper by Hori et al. (Dev. Cell), their ChIP-Seq data revealed significant amounts of CENP-A outside the chicken centromeres Z and 5, but again overall amounts were not given. The authors should expand their discussion to explore this. For example, how much of a non-centromeric pool is there likely to be for other CCAN components? The answer could be similar to that for CENP-A. Thus, the key question is what causes the critical local concentration that nucleates*
*the centromeric chromatin?*

We thank the reviewer for pointing this out, as well as for his enthusiasm regarding our finding. We have now added a reference to the non-centromeric pool of CENP-A described in a recent study by Lacoste et al. (Mol Cell 2014), as well references to Camahort et al. and Shang, Hori, et al. as suggested. In addition, we have included a paragraph in the Discussion regarding the questions raised by the reviewer. In particular we discuss the hypothesis that other centromeric proteins exist at low levels throughout the genome, possibly recruited by non-centromeric CENP-A, and ask how this may influence neocentromere formation. Finally, as suggested by the reviewer, we highlighted that the most novel finding regarding this issue is the surprisingly large proportion of CENP-A that is not centromere bound.

*9) It would not detract from the import of the present work if the authors here mentioned the fact that their numbers are not terribly dissimilar from those estimated by Riberio et al. (PNAS), particularly given that the latter were looking at a different organism*.

The estimate from Ribeiro et al. is stated as follows (direct quote): *“A rough estimate of the amount of CENP-A in the fiber in Fig. 4A can be obtained from the number of localizations (a total of 123). Assuming that, on average, each Dronpa molecule switches approximately three to five times (31), the number of labeled CENP-A molecules is approximately 25 to 40. This value should be taken with caution, as Dronpa can switch as many as 170 times (28). For this estimation, we also assume that most CENP-A in the fibers is labeled with Dronpa and that the amount of endogenous CENP-A is not significant*.*”*

Given the fact that the estimate given in their study is more than an order of magnitude lower than the 400 CENP-A molecules per centromere that we find, that the measurements were made in a cell line from a different organism, and the admitted experimental limitations of their method, we do not feel that it is appropriate to cite this estimate as similar to ours in this section of the manuscript. Nevertheless, we appreciate the effort made by Ribeiro et al., and it is not our intention to disregard their work. Indeed, we do (and did already in the original submission) appropriately cite their CENP-A estimate when introducing the motivation of our own quantifications.

*10) The statement “Previously, it was shown that CENP-A is interspersed with both H3.1 and H3.3 at the centromere (*[11]*;*
[21]*).” should also reference the Ribeiro et al. study*.

Indeed, the Ribeiro et al. study shows that H3 is interspersed with CENP-A at centromeres. However, this finding was originally described in 2002 by Blower et al. Since then, there have been many studies that have confirmed and expanded on this finding, including the Ribeiro study, as well as e.g. Sullivan & Karpen, NSMB 2004 and Sullivan et al., Chrom Res 2011. For the sake of conciseness, we had previously chosen to only reference the original study demonstrating this, as well as Dunleavy et al., which was the first to show that both H3 variants mentioned exist at the centromere. For the sake of completeness, we have now added a reference to the Ribeiro et al. study, as well as the other studies mentioned above.

*11) The authors go on in the same paragraph to say “Nevertheless, this non-centromeric pool only represents <0.1% of all nucleosomes in the genome and thus CENP-A is ∼50-fold enriched (per unit length of DNA) at centromeres (Figure 7a [currently *Figure 8*]). This explains how, despite being outnumbered 25:1 by other H3 variants at the centromere, CENP-A can still accurately specify the centromeric locus.” Are they claiming to offer a convincing explanation of how the CENP-A accurately specifies the centromere? If they really believe this, perhaps they could expand and come up with a more directly testable model*.

We have toned down our conclusion by saying that “it may explain” this. In addition, we have added a suggestion for a testable hypothesis (artificially tethering differential amounts of CENP-A).

*12) Where the authors say “Indeed, it has recently been shown that this sequence specific DNA binding protein has a role in phasing CENP-A nucleosomes (*[36]*)”, they might like to read, and possibly cite, the rather extensive discussion of this in Pluta et al. (JCB 116, 1081, pp. 1091, 1092). The discussion of the enigmatic CENP-B aside, I feel (as already stated above) that the reason for the low levels of CENP-A on the Y chromosome most likely arises from the small size of the alphoid array, which the authors should mention here as well*.

We thank the reviewer for pointing out the rather interesting Pluta et al. paper, of which we were indeed not aware. We have included their hypothesis that CENP-B disrupts the 3D structure of centromeric chromatin in our Discussion. In addition, as outline above (point 7 of this reviewer), we have modified the text to reflect the concern raised regarding the size of the alphoid array.

*13) The authors state “the fixed ratio between total and centromeric CENP-A levels prevents excess CENP-A from accumulating at high density at non-centromeric loci”. I believe that this is an overstatement of what the data actually show. This is still, in my mind, a hypothesis - not a rigorously proven fact, and should be stated as such*.

We have modified the text to clarify that this is indeed a hypothesis

*14) Lastly, I do not understand Figure 7b. What is the “critical size” and why is this bigger than*
*the kinetochore?*

In the model figure (currently Figure 8) we represented CENP-A chromatin as a region of which the size is independent of the downstream centromere complex and kinetochore (as based on data in Figure 6). We show that reduction in centromere CENP-A levels by half results in modest but measurable defects. This suggest that centromeric CENP-A is in excess and the critical amount of CENP-A is less than this, possible hovering around half the full amount or less. We aimed to reflect this finding in the model. We realized however, that “critical size” may not be the right term as changes in CENP-A quantity do not necessarily reflect centromere size (as measured as the part of the genome covered by CENP-A). We have now changed this to “critical quantity”

*Reviewer 3*
*Comments:*

*The authors should discuss possible limitations of their findings, and be careful when making concluding statements. For example, the authors tried to correlate further reduction of CENP-A from the parental RPE line with increases in micronuclei. I am not convinced that the slightly increased frequency of micronuclei is due to a moderate decrease in CENP-A. They would need to show that restoring CENP-A suppresses the phenotype. It is especially a problem when the authors attempt to say that this amount of CENP-A represents the number in “normal” tissue. The results are meaningful in the background of the RPE cell line, but even it is a relevant number, it still has limited value since they have shown that CENP-A levels vary greatly among cell lines and there is no strict correlation with the level of genome instability. This suggests that other 'background' mutations probably contribute greatly to the level of genome instability, and that they only made a weak correlation between CENP-A reduction and increased micronuclei. Overall, these issues raise concerns about using micronucleus frequencies as a direct measure of the impact of altering CENP-A levels, and whether CENP-A reduction accounts for the observed phenotype. The authors determined the critical number based on the observations that HeLa cells with 33% CENP-A are not viable but RPE cells with 40% CENP-A are viable, so they reason that the critical number must lie somewhere in between these two numbers. However, these are completely different lines, with HeLa cells carrying hundreds or thousands of other mutations not in the RPE background. Viability is a combined readout of more than one related pathway. Viability of HeLa cells and RPE cells may have different sensitivity to the levels of CENP-A due to potential mutations in related pathways*.

As outlined in the beginning of this response, we have toned down the conclusions that we make based on the micronucleus results. Furthermore, we agree with the reviewer that the percentages at which CENP-A become critical will vary between cell lines.

However, the objective of this section is to provide a discussion on the theoretical probabilities that centromere function can be lost due to stochastic loss of CENP-A. The percentages of CENP-A occupancy in HeLa and RPE cells are used to guide this discussion. Our main findings that 1) the loss rate of most CENP-A on any centromere is exceedingly low when starting with 200 nucleosomes and 2) that this chance increases dramatically with a smaller CENP-A domain size, still stand. To reflect the concern of the reviewer we have now modified the text to highlight that these numbers are estimates.

*There are several places the authors cite improper literatures, please cite original studies. For example, in introduction, to make the point that CENP-A is so far the primary candidate for centromere identity, the authors need to cite papers beyond just studies on human CENP-A; in the next sentence, to show the point that CENP-A is stably transmitted through cell divisions, there are much more authoritative papers on humans and other organisms than the authors' own papers; the paper cited for primary DNA sequence being neither necessary nor sufficient for centromere determination is Amor et al. in 2004, which is not the best for making this point*.

Original papers have now been included in the above mentioned sections of the paper. Although we have addressed the issue raised by citing more literature on different species, we feel that the relevance of these is limited since our study focusses exclusively on human CENP-A and our findings may not be relevant in all systems. Indeed, in some species (most prominently budding yeast), specific DNA sequences rather than CENP-A nucleosomes are the primary candidate for centromere identity. On the point of CENP-A stability, to our knowledge there are no studies in human cells, other than our own ([44] and [13]), that convincingly demonstrate stable transmission of CENP-A. Indeed, these two studies represent the original finding and most compelling evidence for this, respectively.

*In several places the authors stated results as if they are original observations. However, some of the principles have been revealed or suggested previously, and are here reinforced by the measurements in this manuscript. For example, the correlation between reduction of CENP-A and increased genome instability (if indeed due to CENP-A reduction in the RPE background), and different amounts of CENP-A are at different centromeres. The authors should be careful and need to cite the literature properly*.

We have now clarified in the text that the finding that CENP-A is important for centromere function is well established. However, our finding at which level CENP-A becomes critical is novel. Further, we are aware that others have observed chromosome specific levels for CENP-A which we highlighted and cited in the text in the original submission. For both the Y-centromere and the neocentromere on chromosome 4 we have in fact already cited all appropriate papers in our original submission. We hope this is now more clearly highlighted.

*1) Local background correction for Figure 2 is problematic. The basis of doing local background correction is the assumption that signals immediately surrounding centromeres are noise. What is the factual foundation of this methodology considering that recent papers and the authors' own data indicated the existence of non-centromeric CENP-A in chromatin? If the local background is indeed extremely low, treating it as background noise may not distort the actual number significantly*.

We do not understand the reviewer’s argument that we assume that the surrounding background is exclusively noise-derived. The aim in our experiments is to determine the centromeric levels of CENP-A only. Indeed, the reason that we decided to perform local background subtraction on a per centromere basis, rather than overall background correction for the entire cell, is to get rid of all potential sources of background fluorescence. This includes noise, as well as cellular autofluorescence, randomly distributed non-centromeric CENP-A nucleosomes, and soluble CENP-A.

*2) Figure 1, to overcome the difficulty of different transfer efficiency for untagged or tagged CENP-A, the authors could also use known quantities of purified tagged and untagged CENP-A on the same gel to calibrate tagged CENP-A cell lines*.

This solution would be helpful, however, unfortunately we do not have fluorescently tagged CENP-A at high purity available for this. In addition, we feel that this is not essential given that we are primarily concerned with wildtype CENP-A levels and use the fluorescently-tagged CENP-A cell lines as mere fluorescent reference lines. In addition, we verified these measurements with two additional independent methods that arrive at similar numbers. As such we feel that expanding the quantitative Westerns would be excessive.

*3) What is the cause of the observed different levels of CENP-A for different constructs in Figure 1? Did the authors compare different clones? This may suggest that other mutations occurred during generation of these lines along with changes in CENP-A levels*.

The reduction of CENP-A levels in CA^+/-^ and CA^G/-^ cells is likely due to the fact that these cell lines express CENP-A from a single allele, as was already mentioned in the original submission. However, it is indeed surprising that CENP-A levels are increased in the CA^Y/-^ cell line. Although we do not have a good explanation for this, we hypothesize that this is indeed due to random mutations or some other form of adaptation. We have added this hypothesis to the text. Importantly, since these cell lines are used as fluorescent reference cell lines, we can normalize for these differences and they do not impact on our conclusions.

*4) Figure 1f, this experiment needs to be performed in the RPE lines not in HeLa. In addition, since transient CENP-A overexpression leads to incorporation into non-centromeric chromatin through a DAXX dependent pathway, this must represent a re-equilibrium of two different loading systems that utilize different assembly factors (HJURP and DAXX), and between centromeres and non-centromeric loci. Does DAXX mediated incorporation happen in the various RPE cells? The authors only considered canonical CENP-A loading at the centromere, but I don't think that it makes much sense to talk about mass action without considering the DAXX dependent assembly that may compete with HJURP for centromeric assembly. This will clearly influence the size of the available CENP-A pool for centromeric loading*.

The experiment described in Figure 1 required synchronization of cells in S phase to ensure a single round of CENP-A incorporation. However, drugs that lead to an S phase arrest, such as thymidine can induce p53 dependent apoptosis in RPE cells. Furthermore, upon thymidine release a large proportion of surviving RPE cells are unable to exit the induced S phase arrest and thus synchrony is lost almost immediately. However, p53 deficient HeLa cells can be synchronized using thymidine and we thus opted to use these cells for this particular experiment. An explanation of this has been added to the text.

Very recently, it has been shown that stable overexpression of CENP-A in HeLa cells leads to DAXX mediated incorporation of CENP-A in non-centromeric chromatin (Lacoste et al., Mol Cell 2014). Indeed, we also find that a significant pool of CENP-A is incorporated into non-centromeric chromatin in RPE cells. While this is likely DAXX mediated in RPE cells as well, we do not have any evidence for this. Indeed, the non-centromeric assembly pathway of CENP-A lies well beyond the scope of our study. Nevertheless, the centromeric assembly is most likely still mediated by HJURP, as was observed in CENP-A overexpressing HeLa cells as well (49). The Lacoste et al study was published during the process of review of our manuscript, and thus no mention was made of it originally. We now cite this study in the text and throughout the Discussion.

Given that our data show a strong correlation of CENP-A protein expression levels and centromere incorporation (Figure 1) and that the centromeric fraction is independent of CENP-A expression levels (Figure 2), we feel that we provide sufficient evidence for a mass-action mechanism, irrespective of the specific pathway used to reach this equilibrium.

*5) Figure 4b and 4c, the authors measured fluorescence associated with mitotic centromeres. They found that different centromeres behave differently in the same cell. Using data from multiple cells, they concluded that segregation of CENP-A nucleosomes is random. To conclude this, the authors should have compared the behavior of an individual pair between different cells. It is possible that segregation is non-random for a given centromere of the same chromosome, but varies between centromeres of different chromosomes. Combining data from all centromeres without separating individual chromosomes may give rise to the frequency graph in*
Figure 4*. In addition, the authors need to consider that any differences in the levels of CENP-A at different centromeres/sisters observed in mitosis could be counteracted during new CENP-A assembly in the subsequent G1 phase*.

We appreciate the reviewer’s concern on this point regarding the current Figure 5. However, we would argue that it is unlikely that this is a major issue, as all of the following would need to be true simultaneously: 1) specific centromeres display a non-random CENP-A distribution; 2) the non-randomness is different for different centromeres; and 3) the different degrees of non-randomness between centromeres of different chromosomes add up to be Gaussian in their average distribution. The consistency of our data with random segregation is further emphasized by our finding that different specific centromeres display a similar variance across cells (see Figure 7). Finally, it would be technically challenging, if not impossible, to track sister centromeres of a specific chromosome for the purpose of ruling out this theoretical possibility.

*6) The variation of numbers of CENP-A per centromere using different cell lines is over 2.5 fold, which is fairly large. What could be contributing*
*to this variation?*

The reviewer raises an interesting point here. Although we do not currently have any positive evidence regarding the cause of this variability, we have now excluded the possibility that it is cell cycle dependent (as outlined above and presented in current Figure 3). Although this would be challenging, if at all possible to test, we propose that this reflects inherent variation present between cells.

*7) Figure 5 [current *Figure 6*], the micronuclei phenotype has very limited value in assessing centromere defects. It is actually problematic as the end phenotype for centromere dysfunction, since there could be defects in sister separation (cohesion), as well as genome instability/breakage. It is more direct and informative to look at chromosome segregation. Without additional experiments, e.g. live studies, the results and their implications are hard to interpret. The authors also need to evaluate more than one clone for each line*.

This point is shared by the different reviewers and we agree that interpretation of these results is by necessity fairly limited. We have now extensively modified and subdued our original discussion on this point, as outlined above.

*8) For the phenotype in Figure 5 [current *Figure 6*], how is it striking to see a correlation between reduced CENP-A and increased genome instability? More important, it is not known whether this is due to the problem of the tags or other potential mutations arising during generation of the cell lines. To conclude that the micronuclei phenotype is caused by reduction of CENP-A, the authors would have to do a rescue experiment to restore CENP-A to normal*.

While loss of CENP-A is predictably leading to centromere failure, mitotic defects and ultimately cell death, our finding that a reduction to half the normal levels leads to defects is of interest because it represents the level at which the centromeric CENP-A pool becomes critical. In other words, we do not report here that CENP-A is important for mitosis (which is indeed well established), but we report at what stage it becomes critical. We have now focused our discussion on this point, which we believe is still a valid one, and have attenuated the interpretation of the mechanisms underlying the micronuclei formation.

*9) Figure 6, the authors made a statement about the variance of CENP-A between untransformed cells and cancer cells based on few lines from each category. This statement should be removed. It is safe to say that variance in cancer cell lines is big (6-fold range), but I don't think the evidence is sufficient to say anything about untransformed cells*.

We modified this statement regarding the current Figure 7, and simply made the point that the numbers we find in RPE cells are similar to those in primary cells.

*10) In the Discussion, please rephrase “the fixed ratio between total and centromeric CENP-A levels prevents excess CENP-A from accumulating at high density at non-centromeric loci, thus further reducing the probability of neocentromere formation.” Fixed ratio is an observation, it should not be taken as a mechanism to explain how neo-centromeres are prevented*.

As mentioned above, this sentence has been revised to state “may prevent” rather than “prevents” to indicate that this is a hypothesis.

*11) Also the model figure shows widely spaced single CENPA nucleosomes and could easily be clustered more*.

The reviewer is correct in this assessment, and this alternative possibility was added to the figure legend.